# A versatile platform for chemical engineering of exosomes empowered by ADP-ribosyl cyclases

Lei Zhang [1,6], Srinivasarao Singireddi[1,6], Arshad J. Ansari [1], Guoyun Kao[1], Sunny H. Kim[1], Zeyu Zhang[1], Kaiyu Shen[1], Thuc Oanh Hoang [1], Xinping Duan[1], Qinqin Cheng[1], Tautis Skorka[2] & Yong Zhang [1,3,4,5] ✉

Cell-secreted exosomes have been emerging as an increasingly attractive form of nanomaterials for biomedical research. Various approaches have been established to genetically modify exosomes with proteins of interest for new and/or improved functions. However, equipping exosomes with diverse non-protein biomolecules remains largely dependent on random chemical conjugation or membrane insertion, hindering the application potential of exosomes. Herein, we develop a technology for site-specific functionalization of exosome with different synthetic groups by exploiting surface-expressed CD38, an ADP-ribosyl cyclase, and its covalent inhibitor derived from nicotinamide adenine dinucleotide (NAD[+]). The designed ADP-ribosyl cyclase-enabled exosomes (ARC Exos) carrying conjugated fluorescent imaging probes, small-molecule ligands, cytotoxic payloads, and bone-targeting agents are demonstrated with in vitro and/or in vivo activities and specificities. This ARC Exos-based platform provides a general approach with great versatility for chemically reprogramming exosomes.

Exosomes are nanoscale cellular vesicles known for participating in cell-to-cell communication[1–4]. Among various forms of natural and synthetic nanoparticles, exosomes are characterized by abundant proteins on lipid bilayer membrane and diverse cargo molecules in distinct sizes and structures from parental cells[5–7]. Given their potential in modulating functions of recipient cells via cargos delivery and/or membrane proteins engagements as well as high biocompatibility, exosomes have attracted remarkable interest for therapeutic development[8–11].

In addition to harnessing native exosomes with varied cell origins for treatment of different diseases, considerable efforts are devoted to engineer these cell-derived nanovesicles[12–16]. Notably, a variety of genetic approaches have been established to arm exosomes with new functions, properties, and specificities through expressing exogenous proteins such as antibodies[17,18], enzymes[19,20], ligands[21,22], receptors[23,24], and others in exosomes[25–28]. Such genetically modified exosomes can selectively bind to target cells, efficiently deliver functional cargos to acceptor cells, and effectively modulate pathophysiology of diseased tissues, providing therapeutic candidates with promising pharmacological activities. Despite numerous successes, genetic engineering is largely limited to expression of protein molecules of interest.

To modify exosomes with a variety of non-protein biomolecules, chemical conjugations are often required. Current methods for chemically modifying exosomes mainly involve non-specific reactions with side chains of surface lysine[29,30] or cysteine residues[31,32] or random membrane insertion of lipid conjugates[33–36]. These undefined

[1]Department of Pharmacology and Pharmaceutical Sciences, Alfred E. Mann School of Pharmacy and Pharmaceutical Sciences, University of Southern California, Los Angeles, CA, USA. [2]Molecular Imaging Center, Department of Radiology, Keck School of Medicine, University of Southern California, Los Angeles, CA, USA. [3]Department of Chemistry, Dornsife College of Letters, Arts and Sciences, University of Southern California, Los Angeles, CA, USA. [4]Norris Comprehensive Cancer Center, University of Southern California, Los Angeles, CA, USA. [5]Research Center for Liver Diseases, University of Southern California, Los Angeles, CA, USA. [6]These authors contributed equally: Lei Zhang, Srinivasarao Singireddi. ✉e-mail: yongz@usc.edu

strategies could not only produce heterogeneously modified exosomes, but also adversely impact structures and functions of exosomal proteins and membranes.

As a member of the ADP-ribosyl cyclase family, CD38 membrane protein features an extracellular domain that can robustly convert nicotinamide adenine dinucleotide (NAD+) into cyclic ADP-ribose and ADP-ribose (ADPR)[37–45]. 2′-Cl-arabinose NAD+ (2′-Cl-araNAD+) is a potent covalent inhibitor of CD38 (Fig. 1)[46]. Following CD38-catalyzed release of the nicotinamide leaving group, the resulting ADP-2′-Cl-arabinose can rapidly form a stable arabinosyl-ester bond with the side chain of the glutamate 226 (E226) residue at CD38 active site[46].

Here, we develop an approach for chemical engineering of exosomes in a site-specific fashion. This is achieved by exploiting enzymatic activity of CD38 displayed on exosome surface (Fig. 1). Through 2′-Cl-araNAD+-mediated covalent attachments, catalytically active CD38 expressed on exosomes via a designer fusion protein can be site-specifically functionalized with small-molecule payloads, including Alexa 488 fluorophore, folic acid (FA) ligand, monomethyl auristatin F (MMAF) tubulin inhibitor, and bisphosphonate (BP), a bone-targeting agent. These synthetically functionalized exosomes, namely ADP-ribosyl cyclase-enabled exosomes (ARC Exos), show unique biological activities, supporting a general approach for chemical engineering of exosomes.

## Results

### Design of exosomes with surface-anchored catalytically active CD38

Considering our previous discovery that 2′-Cl-araNAD+ can sustain inhibition of CD38 through a stable covalent bond with the catalytic E226 residue[46], we envisioned that expression of the CD38 enzymatic domain on exosomes may facilitate site-specific chemical conjugations by coupling with 2′-Cl-araNAD+ carrying functional groups at the adenine N6 position (Fig. 1). To this end, we attempted to display enzymatically active CD38 on exosomes by genetically fusing its extracellular domain to N-terminus of human platelet-derived growth

factor receptor (PDGFR) transmembrane domain (TMD) or full-length CD38 to C-terminus of human CD9 (Fig. 2A, and Supplementary Table 1). Both PDGFR TMD and CD9 were known fusion partners to promote expression of proteins on exosomes[17,19]. Unlike the TMD of the type I transmembrane protein PDGFR requiring N-terminal fusion of the CD38 extracellular domain for surface display, the tetraspanin CD9 featuring intracellular N- and C-termini may help express the CD38 enzymatic domain on exosomal surface through a C-terminal fusion with the full-length type II transmembrane protein CD38. A flexible G4S or (G4S)2 linker was inserted between CD38 and PDGFR TMD or CD9 and CD38, respectively, to improve folding of the designed CD38 fusion proteins as well as separation of the CD38 catalytic domain from the fusion partner. In addition, a hemagglutinin (HA) tag was placed at N-terminus for each fusion protein.

Following transient transfection of Expi293F cells with mammalian expression vectors encoding the designed fusions, CD38-PDGFR exosomes and CD9-CD38 exosomes were purified via differential centrifugation. The yields for both types of exosomes are around 2.1 mg per liter of cell culture, comparable to that of native exosomes. Immunoblot analyses indicated expression of CD38-PDGFR TMD and CD9-CD38 fusion proteins in exosomes as revealed by an anti-HA antibody and an anti-CD38 antibody and CD9, CD81, and CD63 markers for all purified exosomes (Fig. 2B). The sizes of CD38-PDGFR TMD and CD9-CD38 fusion proteins are consistent with molecular designs.

Next, nicotinamide guanine dinucleotide (NGD+)-based fluorescence assays were carried out to assess enzymatic activities of CD38 expressed on exosomes. As a substrate of CD38 enzyme, NGD+ can be converted to fluorescent cyclic GDP-ribose[47]. Unlike native exosomes without detectable fluorescence changes in the presence of NGD+, both CD38-PDGFR exosomes and CD9-CD38 exosomes revealed time-dependent fluorescence increases upon mixing with NGD+ as the recombinant CD38 extracellular domain (Fig. 2C), supporting enzymatically active CD38 displayed on exosome surface. Importantly, CD9-CD38 exosomes showed significantly higher catalytic activity than CD38-PDGFR exosomes based on fluorescence intensity (Fig. 2C).

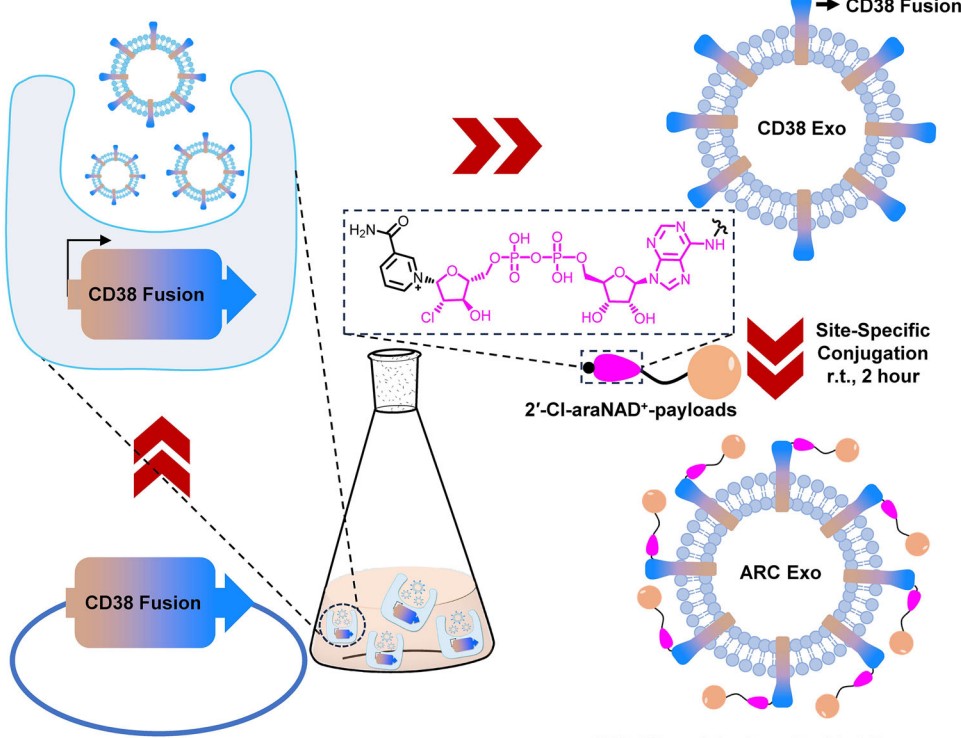

**Fig. 1 | Schematic of the development of site-specifically functionalized ARC Exos by utilizing surface-anchored CD38 enzyme and its covalent inhibitor 2′-Cl-araNAD+.**

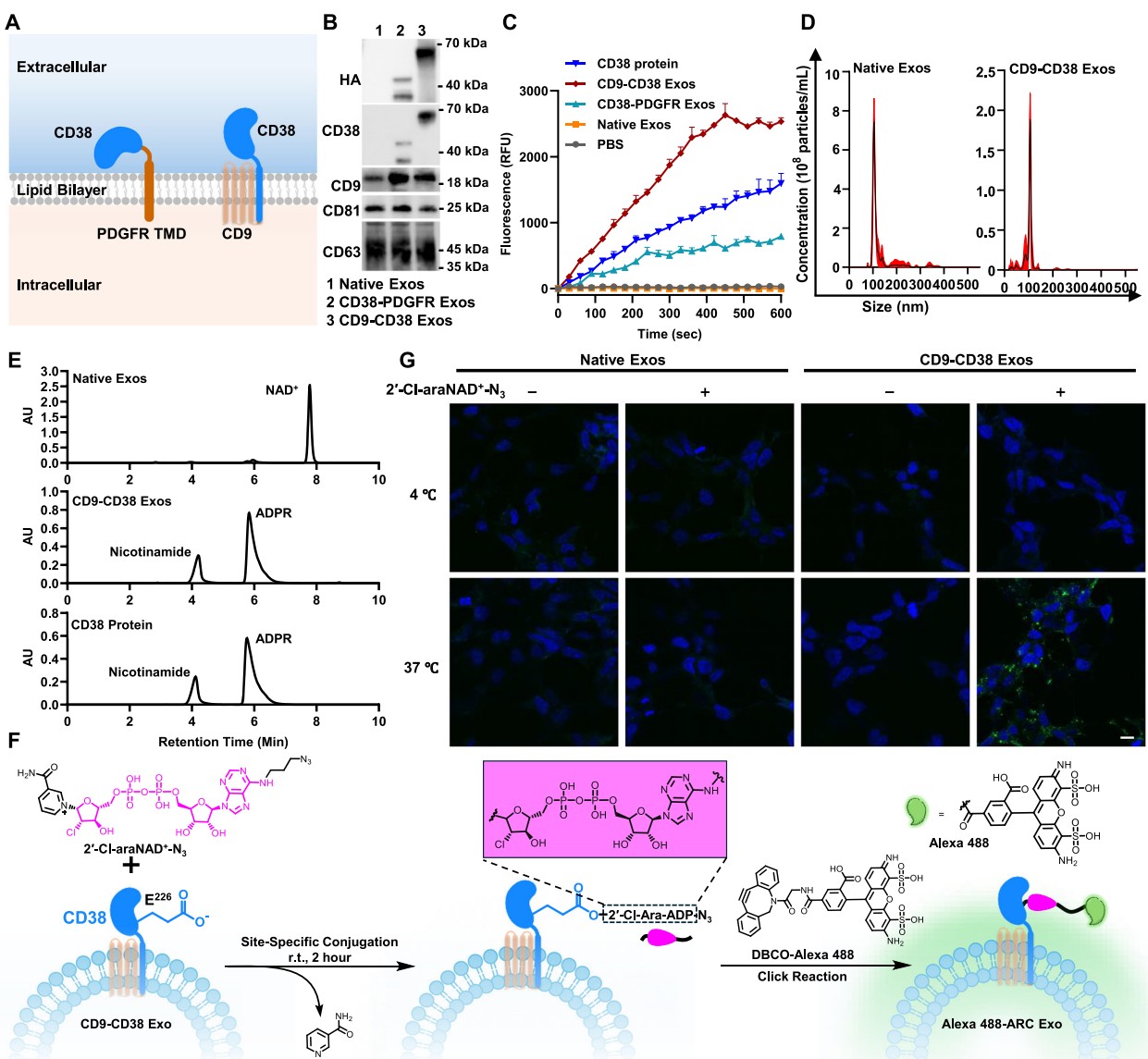

**Fig. 2 | Design of CD38-expressing exosomes and fluorophore-conjugated ARC Exos. A** Schematic representation of designed CD38 fusion proteins for display on exosomal surface via fusing the CD38 extracellular domain to N-terminus of PDGFR TMD (left) or full-length CD38 to C-terminus of CD9 (right). **B** Immunoblot analysis of purified exosomes. **C** Enzymatic activity of CD38-expressing exosomes with NGD⁺ substrate. Exosomes (10 μg mL⁻¹) or CD38 protein (20 nM) were incubated with NGD⁺ (100 μM) in PBS. Formation of fluorescent cGDPR product was monitored by fluorescence signals at 410 nm (*n* = 3 per group). Three biological replications were performed. Data are shown as mean ± SD. **D** NTA analysis of purified native exosomes and CD9-CD38 Exos (*n* = 6 per group). Data are presented as mean ± SD. **E** HPLC analysis of catalytic activity of CD9-CD38 Exos with NAD⁺ substrate. Exosomes (50 μg mL⁻¹) or CD38 protein (200 nM) were incubated with NAD⁺ (200 μM) for 4 h and then analyzed by HPLC. **F** Synthesis of Alexa 488-ARC Exos. **G** Visualization of exosomes uptake by HEK293T cells. CD9-CD38 exosomes and native exosomes were incubated without or with 2′-Cl-araNAD⁺-N₃, followed by click reactions with DBCO-Alexa 488. The resulting Alexa 488-ARC Exos along with other exosome controls (50 μg mL⁻¹) were incubated with HEK293T cells for 2 h on ice or at 37 °C, followed by washing, fixation, and confocal microscopic analysis. Scale bars: 20 μm. Experiments in **B**, **G** were repeated independently three times with similar results. Source data are provided as a Source Data file.

Nanoparticle tracking analysis (NTA) indicated similar size distribution for native exosomes, CD38-PDGFR exosomes, and CD9-CD38 exosomes (Fig. 2D, and Supplementary Fig. 1a). Based on the measured ADP-ribosyl cyclase activities of recombinant CD38 extracellular domain, CD9-CD38 exosomes, and CD38-PDGFR exosomes by NGD⁺ assays and nanoparticle concentrations (Fig. 2C, D, and Supplementary Fig. 1a), it was estimated that on average, each CD9-CD38 exosome vesicle carries ~1370 surface-displayed CD38 molecules and each CD38-PDGFR exosome particle contains around 167 CD38 molecules on surfaces (Supplementary Fig. 1b). The numbers of CD38 on CD9-CD38 exosomes and CD38-PDGFR exosomes quantified by NGD⁺-based enzymatic activity assays are consistent with those determined

by immunoblot and ELISA analyses (Supplementary Fig. 1c–e). These data indicated elevated levels of CD38 expression on CD9-CD38 exosomes relative those of CD38-PDGFR exosomes and a positive correlation between CD38 expression and its catalytic activity on exosomes (Fig. 2C). Therefore, CD9-CD38 exosomes were selected for conjugation with 2′-Cl-araNAD⁺ analogues carrying different functional groups to produce ARC Exos.

Furthermore, the enzymatic activity of CD9-CD38 exosomes was verified by high-performance liquid chromatography (HPLC) (Fig. 2E, and Supplementary Fig. 2a–d). In contrast to native exosomes lacking catalytic activity for NAD⁺, CD9-CD38 exosomes like recombinant CD38 extracellular domain can catalyze rapid production of

nicotinamide and ADPR from $NAD^+$ under the same conditions. These results indicate successful expression of enzymatically active CD38 on surface of CD9-CD38 exosomes.

## ARC Exos with conjugated Alexa 488 fluorophores
To determine whether CD9-CD38 exosomes allow site-specific functionalization on surface, clickable dibenzocyclooctyne (DBCO)-Alexa 488 was first selected as a model imaging reagent for conjugation (Fig. 2F). CD9-CD38 exosomes were incubated with the synthesized 2′-Cl-araNAD$^+$ with an azido group at the adenine N6 position (2′-Cl-araNAD$^+$-N$_3$), a previously developed covalent inhibitor of CD38[46]. According to the NGD$^+$-based fluorescence assays, 2-h incubation with 2′-Cl-araNAD$^+$-N$_3$ at room temperature (r.t.) in phosphate-buffered saline (PBS) led to full inhibition of CD38 enzymatic activity on exosomes (Supplementary Fig. 2e). DBCO-Alexa 488 dye was subsequently added for click reactions, producing Alexa 488-ARC Exos. CD9-CD38 exosomes incubated with DBCO-Alexa 488 in the absence of 2′-Cl-araNAD$^+$-N$_3$ along with native exosomes labeled by DBCO-Alexa 488 without and with 2′-Cl-araNAD$^+$-N$_3$ under the same conditions were included as controls. Confocal microscopy of HEK293T cells treated with these exosomes revealed cellular uptake of Alexa 488-ARC Exos at 37 °C by green fluorescence (Fig. 2G). In contrast, no significant fluorescence signals were observed for cells incubated with control exosomes. Moreover, nanoflow cytometry revealed that in contrast to lipophilic PKH67 dye that randomly stains exosome membranes, DBCO-Alexa 488 coupled with 2′-Cl-araNAD$^+$-N$_3$ results in more efficient labeling of exosomes and little effects on size distribution of exosomes (Supplementary Fig. 2f, g). These results demonstrate site-specific and homogeneous labeling of exosomes with Alexa 488 fluorescent dyes through surface-expressed CD38 and its covalent inhibitor 2′-Cl-araNAD$^+$-N$_3$.

## Design and analysis of ARC Exos with displayed FA ligands and MMAF payloads
FA, a high-affinity ligand for folate receptor alpha (FRα) frequently overexpressed on cancer cells[48,49], was next attempted for site-specific conjugation with CD9-CD38 exosomes. Following 2-h incubation of the synthesized 2′-Cl-araNAD$^+$-FA conjugate with CD9-CD38 exosomes at r.t., FA-ARC Exos were generated (Fig. 3A, and Supplementary Fig. 3a). ELISA analysis indicated strong binding for FA-ARC Exos toward recombinant human FRα, but little interactions with FRα for native exosomes or CD9-CD38 exosomes (Fig. 3B). Different cancer cell lines with varied expression levels of FRα, including SKOV3 (FRα$^+$), A2780 (FRα$^+$), CAKI-1 (FRα$^-$), and A549 (FRα$^-$) (Fig. 3C), were used to evaluate binding specificity of FA-ARC Exos. After staining of the cells with PKH67-labeled exosomes on ice, confocal microscopy showed significant green fluorescent signals for FRα-expressing SKOV3 and A2780 cells incubated with FA-ARC Exos (Fig. 3D). In contrast, FA-ARC Exos-treated FRα-negative CAKI-1 and A549 cells revealed no green fluorescence. Furthermore, no green fluorescent signals were detected for these cells stained by native exosomes or CD9-CD38 exosomes. These results support functional display of FA ligands on exosomes via 2′-Cl-araNAD$^+$-mediated attachments to surface CD38.

In addition to FA, a tubulin inhibitor MMAF was covalently linked to 2′-Cl-araNAD$^+$[50,51]. The resulting 2′-Cl-araNAD$^+$-MMAF alone or mixed with 2′-Cl-araNAD$^+$-FA at a molar ratio of 1:1 was incubated with CD9-CD38 exosomes to generate MMAF-ARC Exos and FA-MMAF-ARC Exos (Fig. 3E, F, and Supplementary Fig. 3b, c). As revealed by ELISA assays, FA-MMAF-ARC Exos and FA-ARC Exos display comparable binding affinity to human recombinant FRα, whereas MMAF-ARC Exos show no detectable binding at concentrations up to 50 μg mL$^{-1}$, similar to those of native exosomes and CD9-CD38 exosomes (Supplementary Fig. 3d). Importantly, in vitro cytotoxicity assays indicated that when treating FRα-positive SKOV3 and A2780 cells, the potencies of FA-MMAF-ARC Exos increase by approximately 16- and 9-fold relative to those of

MMAF-ARC Exos (EC$_{50}$: 238.5 vs 3877.3 ng mL$^{-1}$ for SKOV3 and 693.9 vs 6499.7 ng mL$^{-1}$ for A2780), respectively (Fig. 3G, H). In comparison, FA-MMAF-ARC Exos and MMAF-ARC Exos have comparable cytotoxicity for CAKI-1 and A549 cells that lack surface expression of FRα (Fig. 3I, J). No cytotoxic effects on these cancer cell lines were observed under the same conditions for FA-ARC Exos, CD9-CD38 exosomes, and native exosomes.

Confocal microscopy indicated that in comparison to MMAF-ARC Exos, FA-MMAF-ARC Exos display significantly increased uptake efficiency for FRα-expressing SKOV-3 and A2780 cells, but not for FRα-negative CAKI-1 and A549 cells (Supplementary Fig. 3e–h). Additionally, quantitative analysis by ELISA showed ~950 MMAF molecules on each MMAF-ARC Exo particle and 490 MMAF molecules on each FA-MMAF-ARC Exo vesicle (Supplementary Fig. 4a, b). The half-life of MMAF-ARC Exos in mouse plasma determined by a MMAF-specific antibody is 43.5 ± 5.0 h (Supplementary Fig. 4c, d). Altogether, these data suggest that the conjugated FA compounds on exosomes promote targeted delivery of cytotoxic MMAF into FRα-expressing cells, supporting generation of exosomes with surface displayed small-molecule ligands and therapeutic payloads.

## Synthesis and assessment of ARC Exos functionalized with bone-targeting BP
To establish ARC Exos as a general approach, bone-targeting small-molecule BP was chosen for site-specific conjugation with exosomes[52–54]. 2′-Cl-araNAD$^+$-BP was first synthesized chemically, followed by one-step 2-h enzymatic reactions with CD9-CD38 exosomes to afford BP-ARC Exos (Fig. 4A, and Supplementary Fig. 5). To evaluate binding activity for bones, BP-ARC Exos along with native exosomes and CD9-CD38 exosomes were labeled with PKH67 dyes and then incubated with non-decalcified bone sections from mice for 1 h at 37 °C. Bone samples were next stained with xylenol orange (XO). Confocal microscopic analysis indicated binding of BP-ARC Exos to bones, but lack of green fluorescence signals on bones for CD9-CD38 exosomes and native exosomes (Fig. 4B). Binding kinetics showed that BP-ARC Exos could reach binding levels of about 90% with hydroxyapatite (HAp) and 80% with bone tissues within 4- and 12-h incubation, respectively (Fig. 4C, D). By contrast, only 20% or lower CD9-CD38 exosomes and native exosomes were bound to HAp or bones under the same conditions. These results support the functionalization of exosomes with bone-specific BP agents by the ARC Exos method.

## Design and characterization of bone-targeted ARC Exos for cancer immunotherapy
To demonstrate the utility of BP-ARC Exos, we co-expressed CD38 enzymes and antibodies against human CD3 and epidermal growth factor receptor (EGFR) on exosome surface by co-transfecting Expi293F cells with CD9-CD38 fusion construct along with a previously created plasmid encoding αCD3 single-chain variable fragment (scFv)-αEGFR scFv-PDGFR TMD fusion (Supplementary Fig. 6)[17]. The resulting CD9-CD38/αCD3-αEGFR exosomes (Exos) were next incubated with the synthesized 2′-Cl-araNAD$^+$-BP conjugate to generate BP-αCD3-αEGFR-ARC Exos that are expected to recruit and activate cytotoxic T cells toward elimination of bone-metastasized EGFR$^+$ tumor cells via surface-anchored αCD3 and αEGFR antibodies and BP molecules (Fig. 5A, and Supplementary Fig. 7a). As the bone is one of the most frequent places for cancer metastasis[55–57], bone-targeting immunotherapeutic exosomes may result in more effective and safer treatment for metastatic cancer.

Immunoblots support co-expression of the HA-tagged CD9-CD38 fusion and αCD3 scFv-αEGFR scFv-PDGFR TMD fusion proteins in CD9-CD38/αCD3-αEGFR Exos (Fig. 5B). NTA indicated diameters peaking around 100 nm for CD9-CD38/αCD3-αEGFR Exos and BP-αCD3-αEGFR-ARC Exos (Fig. 5C, D), comparable to those of native exosomes and CD9-CD38 exosomes (Fig. 2D). Conjugation of BP

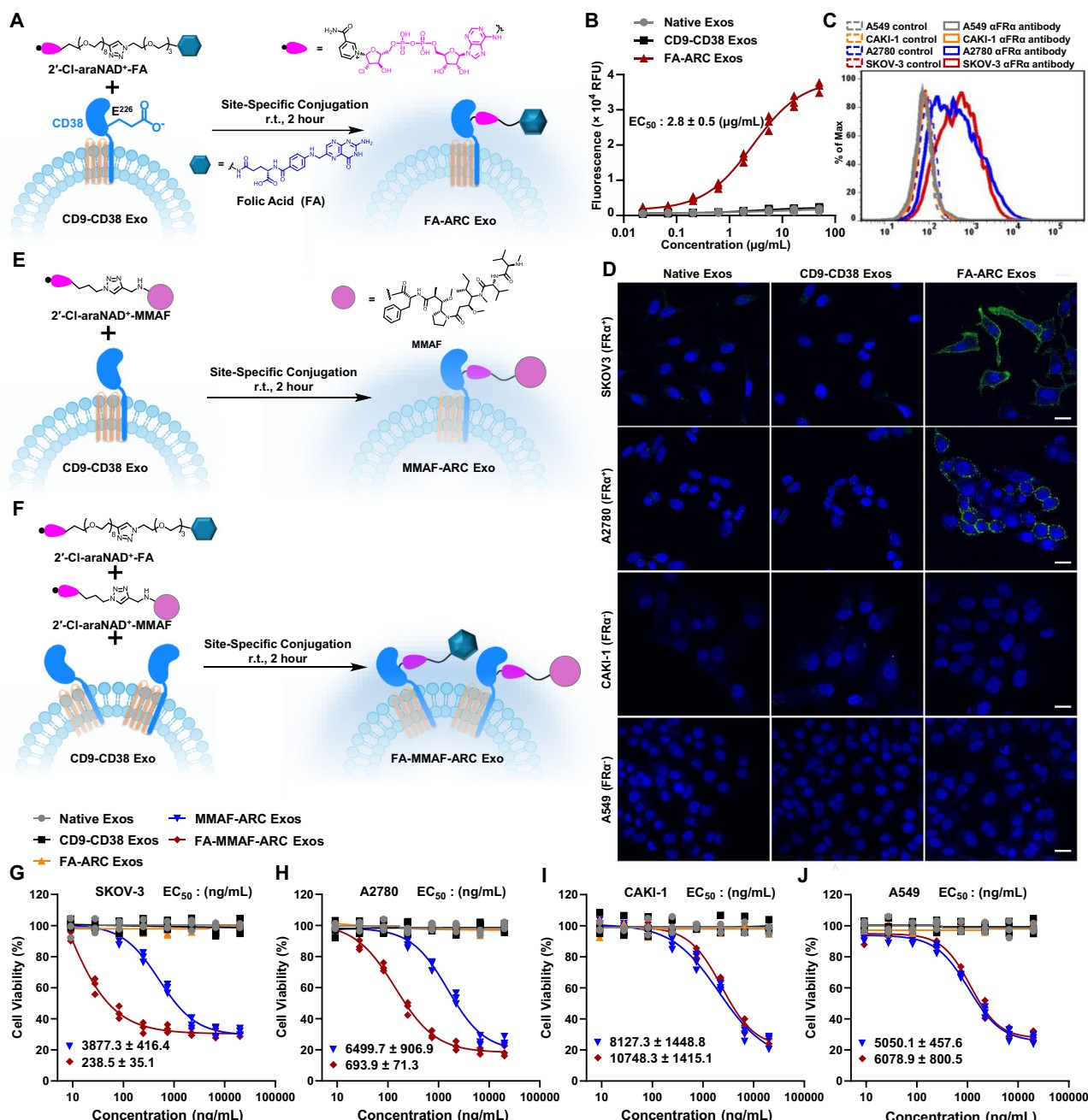

**Fig. 3 | Synthesis and in vitro analysis of ARC Exos with surface-displayed small-molecule ligands and cytotoxic payloads. A** Synthesis of FA-ARC Exos. **B** Sandwich ELISA analysis of binding of FA-ARC Exos to recombinant human FRα. Exosomes and anti-His$_6$ antibody were used as capture and detection reagents, respectively, ($n = 3$ per group). Data are shown as mean ± SD. **C** Flow cytometry analysis of expression levels of FRα on A549, CAKI-1, A2780, and SKOV-3 cells. **D** Binding of FA-ARC Exos to human FRα-positive cancer cells. PKH67-labeled exosomes (50 μg mL$^{-1}$) were incubated with SKOV-3 (FRα$^+$), A2780 (FRα$^+$), CAKI-1 (FRα$^-$), and A549 (FRα$^-$) cells for 1 h on ice, followed by washing, fixation, and confocal microscopic analysis. Scale bars: 20 μm. Experiments in **D** were repeated independently three times with similar results. **E** Synthesis of MMAF-ARC Exos. **F** Synthesis of FA-MMAF-ARC Exos. CD9-CD38 Exos were incubated with a mixture of 2′-Cl-araNAD$^+$-FA and 2′-Cl-araNAD$^+$-MMAF (1:1 molar ratio) for 2 h, followed by removals of free 2′-Cl-araNAD$^+$-FA and 2′-Cl-araNAD$^+$-MMAF. In vitro cytotoxicity of FA-MMAF-ARC Exos for SKOV-3 (**G**), A2780 (**H**), CAKI-1 (**I**), and A549 (**J**) cells. Various forms of exosomes at different concentrations were incubated with FRα-positive and -negative cells for 42 h, followed by measurements of cell viabilities ($n = 3$ per group). Three biological replications were performed. Data are presented as mean ± SD. Source data are provided as a Source Data file.

molecules with CD9-CD38/αCD3-αEGFR Exos led to increases in overall binding to HAp and bones after 12-h incubation from ~20% to 80% (Fig. 5E, F). And the binding kinetics to HAp and bones for BP-ARC Exos and BP-αCD3-αEGFR-ARC Exos are at similar levels. Like native exosomes and CD9-CD38 exosomes, αCD3-αEGFR exosomes (Exos) showed around 20% binding to HAp and bones following 24- and 12-h incubation. Consistent with the kinetic analysis, confocal microscopy studies using PKH67-labeled exosomes revealed selective binding to bone sections for BP-ARC Exos and BP-αCD3-αEGFR-ARC Exos (Fig. 5G).

Furthermore, in vivo biodistribution of BP-αCD3-αEGFR-ARC Exos was examined in mice bearing luciferase-labeled EGFR-positive BT-20

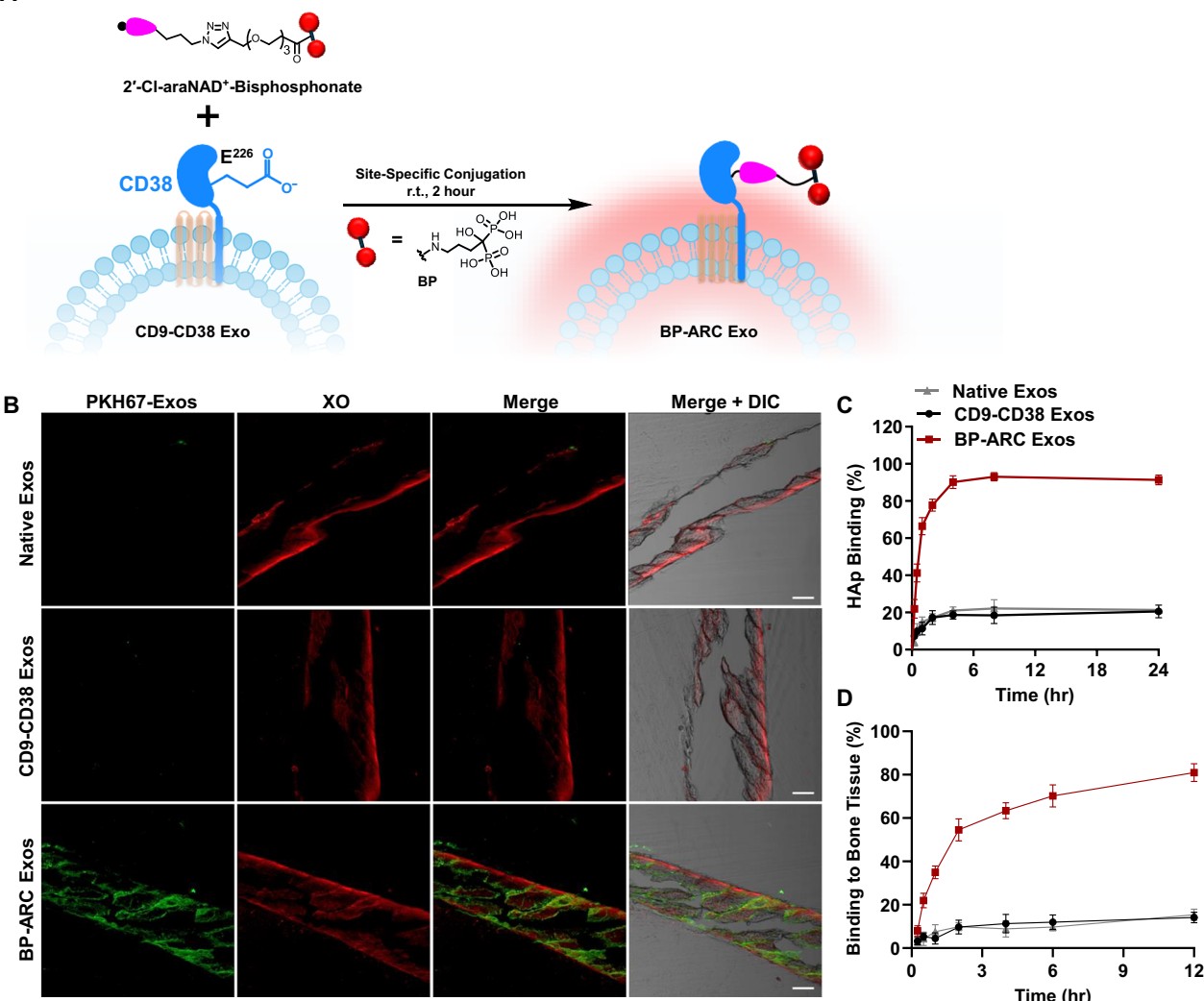

**Fig. 4 | Generation and evaluation of ARC Exos with bone-targeting functionality. A** Synthesis of BP-ARC Exos. **B** Visualization of binding of BP-ARC Exos to bones. Non-decalcified bone sections from mice were incubated with PKH67-labeled exosomes (50 μg mL⁻¹) for 1 h at 37 °C, followed by bone staining with XO (2 μg mL⁻¹) and confocal microscopy. DIC: differential interference contrast. Scale bars: 200 μm. Experiments in Fig. 4**B** were repeated independently three times with similar results. Binding kinetics of BP-ARC Exos to HAp (**C**) and bones (**D**) (n = 3 per group). Three biological replications were performed. Data are presented as mean ± SD. Source data are provided as a Source Data file.

tumors implanted into left hindlimbs via para-tibial injections. Following tumor establishment, mice were intravenously administered with BP-αCD3-αEGFR-ARC Exos or CD9-CD38/αCD3-αEGFR Exos which were labeled by near-infrared cyanine dyes. Fluorescent imaging analysis at 24 h after exosome treatment indicated elevated levels of exosome signals in tumor-bearing hindlimbs for the BP-αCD3-αEGFR-ARC Exo-treated group relative to those of tumor-free hindlimbs of the same mice and hindlimbs without or with tumors from the mice receiving CD9-CD38/αCD3-αEGFR Exos (Fig. 5H). Unlike hindlimbs from CD9-CD38/αCD3-αEGFR Exo-treated mice with weak or undetectable fluorescence intensities at 48–168 h after exosome injections, left hindlimbs with tumors for mice receiving BP-αCD3-αEGFR-ARC Exos showed prolonged fluorescent signals through 168 h, higher than those of tumor-free right hindlimbs from the same animals (Supplementary Fig. 7b). Moreover, fluorescent signals of BP-αCD3-αEGFR-ARC Exos were co-localized with tumor-derived luminescence (Supplementary Fig. 7c). In addition, little or no signals of BP-αCD3-αEGFR-ARC Exos were detected in teeth (Supplementary Fig. 7d). Taken together, these in vitro and in vivo results support the gain of bone-

targeting function for exosomes through site-specific conjugation of BP molecules, promoting enrichment of BP-αCD3-αEGFR-ARC Exos in bones with EGFR-expressing tumors.

## In vitro and in vivo anti-cancer immunity and specificity of BP-conjugated ARC Exos targeting CD3 and EGFR receptors

After verifying the bone-binding specificity of BP-αCD3-αEGFR-ARC Exos, the functions of surface-expressed αCD3 scFv and αEGFR scFv antibodies were validated. Flow cytometric analysis revealed that BP-αCD3-αEGFR-ARC Exos can tightly bind to CD3⁺ Jurkat cells in a fashion similar to αCD3-αEGFR Exos (Supplementary Fig. 8). Binding affinities to Jurkat cells are comparable for CD9-CD38/αCD3-αEGFR Exos and BP-αCD3-αEGFR-ARC Exos. As controls, CD9-CD38 exosomes and BP-ARC Exos without αCD3 scFv antibodies showed no binding to Jurkat cells. Moreover, flow cytometry using breast cancer cell lines with varied expression levels of EGFR indicated that BP-αCD3-αEGFR-ARC Exos along with CD9-CD38/αCD3-αEGFR Exos and αCD3-αEGFR Exos can tightly bind to EGFR-positive BT-20 and MDA-MB-231 cells, but not EGFR-negative MDA-MB-453 cells (Supplementary Fig. 9). In addition,

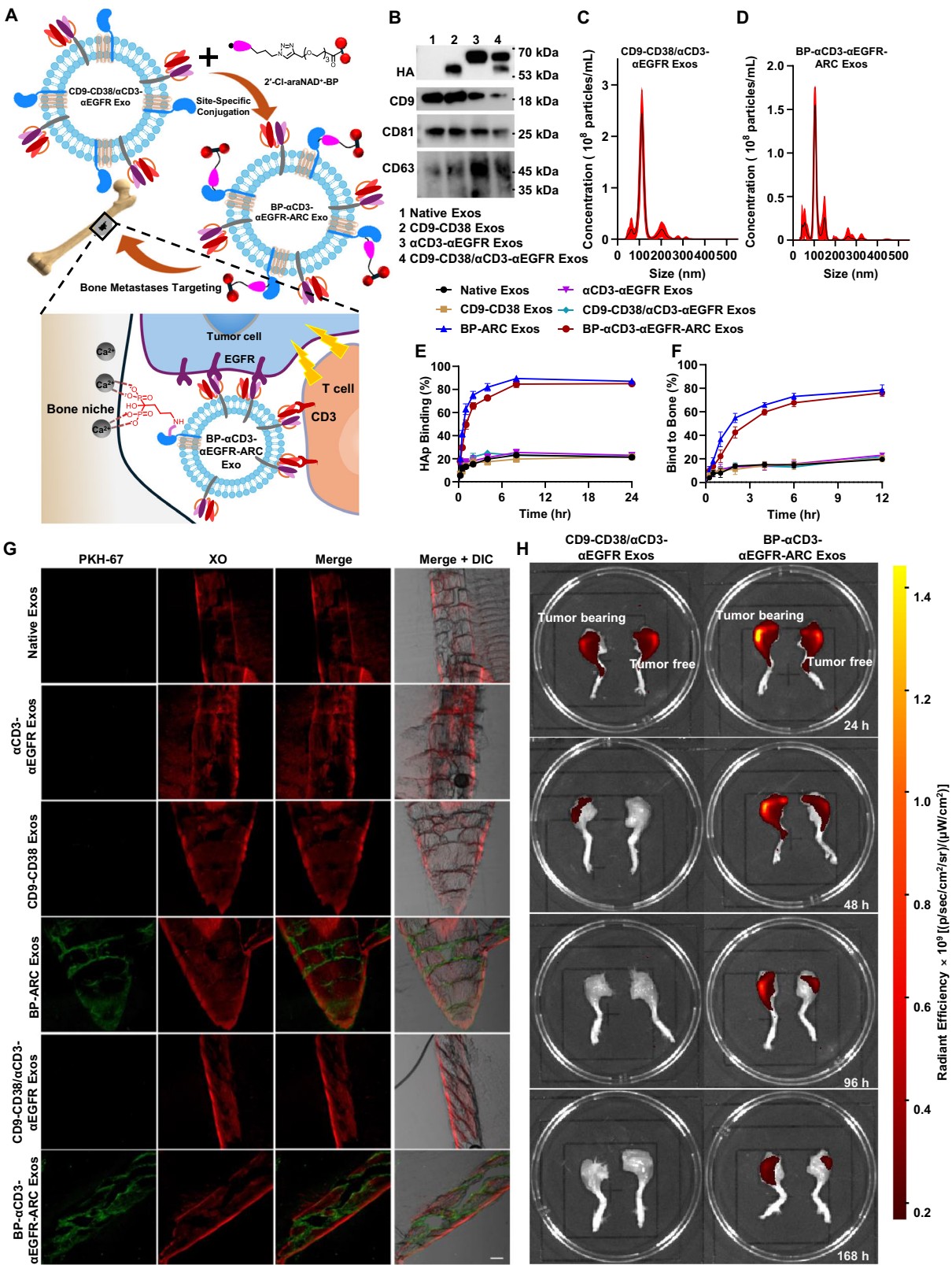

CD9-CD38 exosomes and BP-ARC Exos expressing no αEGFR scFv antibodies lack binding to these cell lines. These data confirm CD3- and EGFR-specific binding activities for BP-αCD3-αEGFR-ARC Exos as well as CD9-CD38/αCD3-αEGFR Exos. Notably, confocal imaging indicated that unlike native exosomes and CD9-CD38/αCD3-αEGFR Exos, BP-αCD3-αEGFR-ARC Exos can promote the clustering of BT-20 cells with T cells on bone tissues (Supplementary Fig. 10a). Flow cytometry

revealed efficient binding of BP-αCD3-αEGFR-ARC to both BT-20 cells and T cells (Supplementary Fig. 10b–e).

To determine the in vitro biological activity of BP-αCD3-αEGFR-ARC Exos that are anticipated to engage T cells to EGFR⁺ tumor cells for killing (Fig. 6A), three types of breast cancer cells including MDA-MB-453 (EGFR⁻), BT20 (EGFR⁺), and MDA-MB-231 (EGFR⁺), were incubated with non-activated human peripheral blood mononuclear cells

**Fig. 5 | Design and characterization of bone-targeted immunomodulatory ARC Exos. A** Schematic of the generation and action mode of BP-αCD3-αEGFR-ARC Exos. **B** Immunoblot analysis of purified exosomes. NTA analysis of CD9-CD38/αCD3-αEGFR Exos (**C**) and BP-αCD3-αEGFR-ARC Exos (**D**) (n = 6 per group). Data are shown as mean ± SD. Binding kinetics of BP-αCD3-αEGFR-ARC Exos to HAp (**E**) and bones (**F**) (n = 3 per group). Three biological replications were performed. Data are presented as mean ± SD. **G** Visualization of binding of BP-αCD3-αEGFR-ARC Exos to bones. Non-decalcified bone sections from mice were incubated with

PKH67-labeled exosomes (50 μg mL⁻¹) for 1 h at 37 °C, followed by bone staining with XO (2 μg mL⁻¹) and confocal imaging. Scale bars: 200 μm. **H** Biodistribution of BP-αCD3-αEGFR-ARC Exos in hindlimbs of mice bearing EGFR-positive BT-20 tumors. NSG mice with BT-20 tumors implanted in left hindlimbs via para-tibial injections were intravenously administered with DiR-labeled exosomes (10 mg kg⁻¹), followed by fluorescence imaging of hindlimbs at 24–168 h. Experiments in **B**, **G** were repeated independently three times with similar results. Source data are provided as a Source Data file.

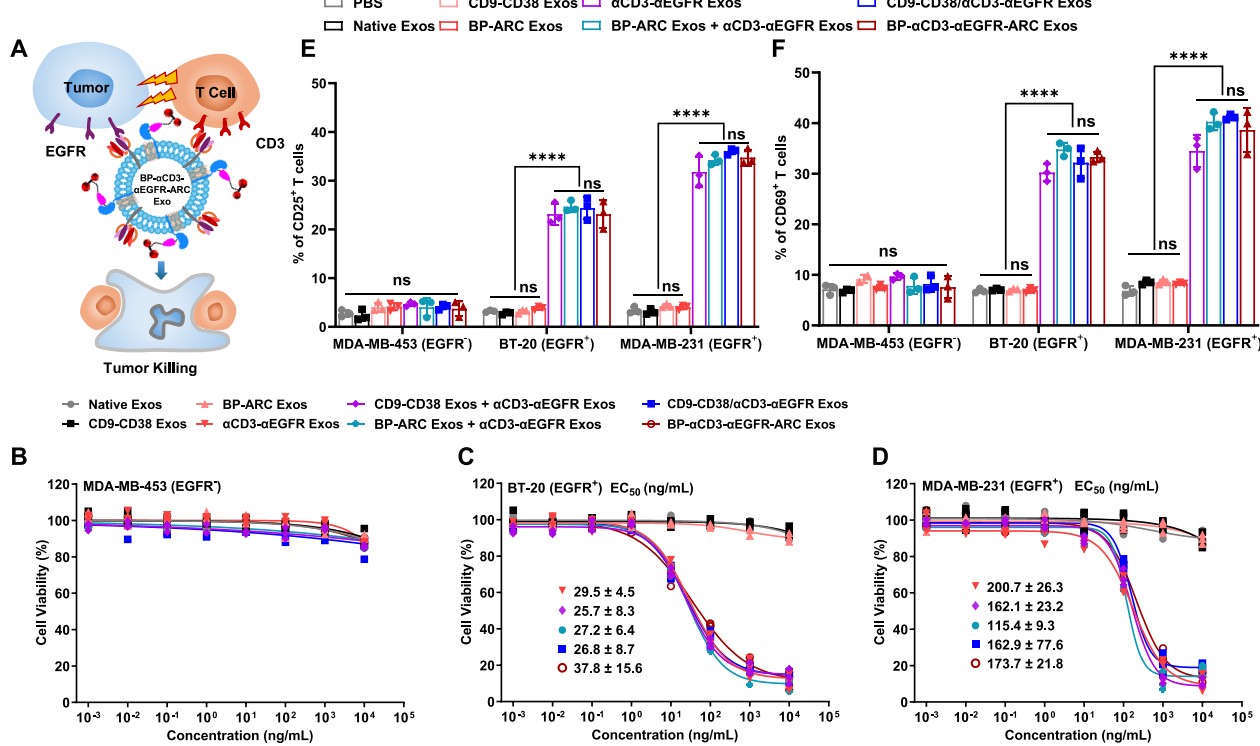

**Fig. 6 | In vitro biological activities of BP-αCD3-αEGFR-ARC Exos. A** Schematic of BP-αCD3-αEGFR-ARC Exos-mediated T-cell activation and attack against EGFR-expressing cancer cells. In vitro cytotoxicity of BP-αCD3-αEGFR-ARC Exos for MDA-MB-453 (EGFR⁻) (**B**), BT-20 (EGFR⁺) (**C**), and MDA-MB-231 (EGFR⁺) (**D**) cells (n = 3 per group). Three biological replications were performed. Data are shown as mean ± SD. EGFR-dependent activation of human T cells by BP-αCD3-αEGFR-ARC Exos as analyzed via T-cell activation markers CD25 (**E**) and CD69 (**F**) with flow cytometry. Non-activated human PBMCs (effector cells) were incubated with cancer cells

(target cells) (E:T ratio: 8) in the presence of PBS or exosomes at various concentrations for 48 h, followed by collection of human PBMC suspensions, determination of cancer cell viabilities, and analysis of CD25- and CD69-positive T cells (n = 3 per group). Three biological replications were performed. Data are presented as mean ± SD. Statistical analysis was performed using ordinary one-way ANOVA with Tukey's multiple comparison test. Significance of finding was defined as follows, ns = not significant p > 0.05 and ****p < 0.0001. Source data are provided as a Source Data file.

(PBMCs) in the presence of purified exosomes at varied concentrations. Viability measurements of target cancer cells showed that BP-αCD3-αEGFR-ARC Exos can selectively induce cytotoxicity against EGFR-expressing BT-20 and MDA-MB-231 cells with EC₅₀ in a range of 37–174 ng mL⁻¹, but spare MDA-MB-453 cells lacking EGFR expression (Fig. 6B–D). In comparison with αCD3-αEGFR Exos, no significant changes in potency were observed for BP-αCD3-αEGFR-ARC Exos, CD9-CD38/αCD3-αEGFR Exos, and combinations (1:1) of αCD3-αEGFR Exos and CD9-CD38 exosomes and BP-ARC Exos and αCD3-αEGFR Exos. Additionally, native exosomes, CD9-CD38 exosomes, and BP-ARC Exos were found to have little or no cytotoxicity with these breast cancer cell lines.

In vitro T-cell activation by BP-αCD3-αEGFR-ARC Exos was also analyzed based on activation markers CD25 and CD69, as well as secreted cytokines interleukin-2 (IL-2) and interferon-gamma (IFN-γ). Flow cytometry indicated significantly higher levels of CD25⁺ and CD69⁺ T cells for EGFR-expressing BT-20 and MDA-MB-231 cells

treated by BP-αCD3-αEGFR-ARC Exos relative to those of cells with PBS, native exosomes, CD9-CD38 exosomes, and BP-ARC Exos (Fig. 6E, F). No obvious increases in percentages of CD25⁺ and CD69⁺ T cells were seen for MDA-MB-453 cells (EGFR⁻) incubated with BP-αCD3-αEGFR-ARC Exos. The population levels of activated T cells were comparable for breast cancer cells treated by BP-αCD3-αEGFR-ARC Exos, αCD3-αEGFR Exos, CD9-CD38/αCD3-αEGFR Exos, and a mixture (1:1) of BP-ARC Exos and αCD3-αEGFR Exos. Moreover, high levels of secreted IL-2 and IFN-γ cytokines were found in BT-20 and MDA-MB-231 cells incubated with BP-αCD3-αEGFR-ARC Exos and other types of exosomes expressing αCD3-αEGFR-PDGFR TMD fusions, consistent with flow cytometric findings (Supplementary Fig. 11). No significant secretions of IL-2 and IFN-γ were observed for EGFR-negative MDA-MB-453 cells in the absence or presence of varied forms of exosomes. In addition, catalytically active CD9-CD38 exosomes show little or weak effects on PBMC viability and T-cell activation (Supplementary Fig. 12). These results indicate that BP-αCD3-αEGFR-ARC Exos can induce

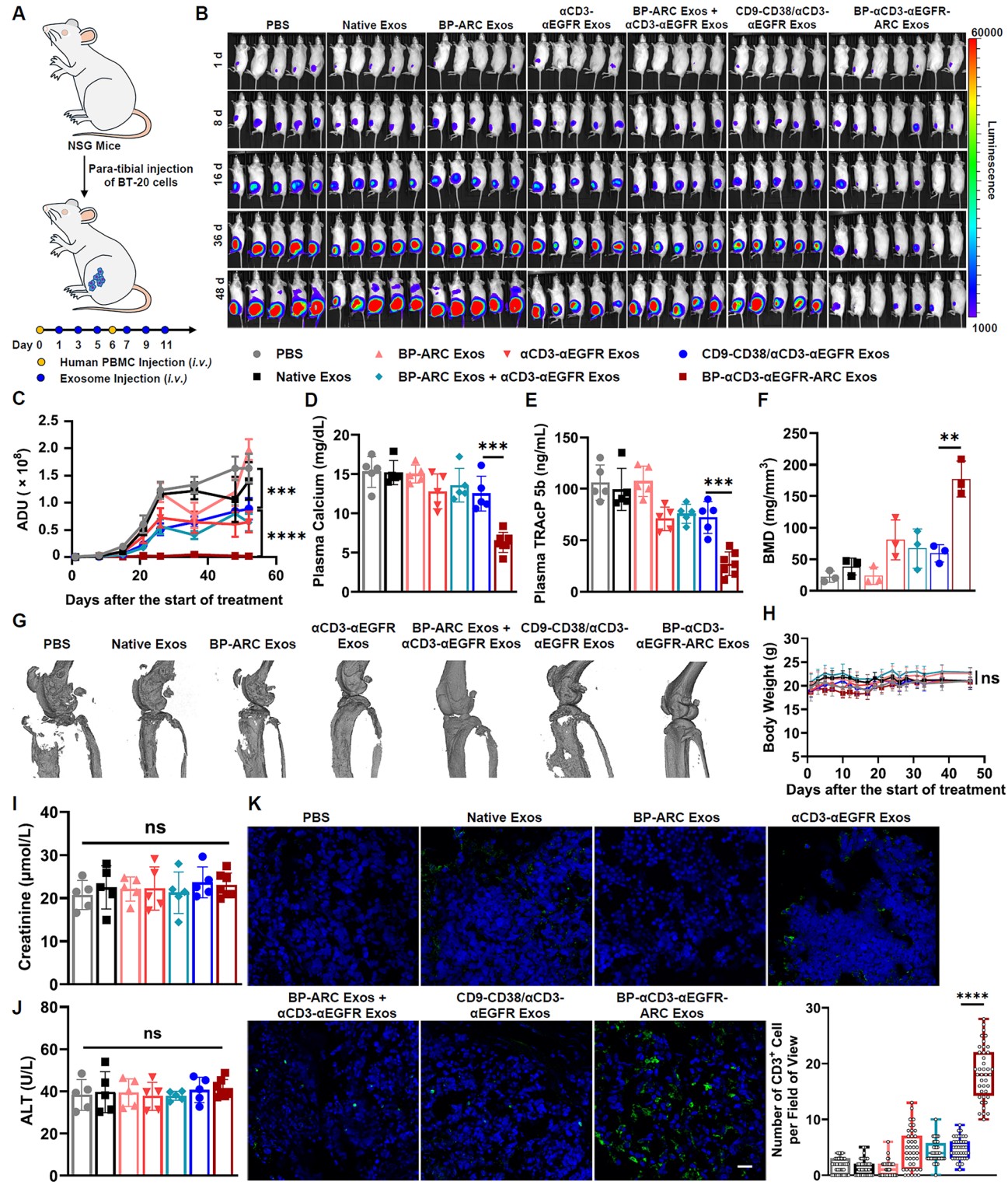

potent killing specific for EGFR-expressing cancer cells through activating cytotoxic T cells. Also, conjugation of bone-targeting BP agents through CD9-CD38 fusion protein has no negative impact on EGFR-dependent cytotoxicity and T-cell activation activity of αCD3-αEGFR Exos.

In vivo therapeutic efficacy of BP-αCD3-αEGFR-ARC Exos was evaluated using a mouse xenograft model of breast cancer. Luciferase-labeled BT-20 cells were inoculated into left hindlimbs of NSG mice via para-tibial injections. Mice were grafted with human PBMCs on days 10 and 16 following tumor implantation

and treated every other day for a total of six times with PBS vehicle or different types of exosomes beginning 1 day after the first PBMC injection (Fig. 7A). Bioluminescence imaging indicate rapid tumor growths on left hindlimbs of mice receiving PBS, native exosomes, and BP-ARC Exos, significantly reduced tumor progression for mice administered with αCD3-αEGFR Exos, CD9-CD38/αCD3-αEGFR Exos, and a combination (1:1) of BP-ARC Exos and αCD3-αEGFR Exos, and remarkable tumor suppression in BP-αCD3-αEGFR-ARC Exos-treated mice (Fig. 7B, C, and Supplementary Fig. 13). Furthermore, plasma levels of bone resorption

**Fig. 7 | In vivo anti-tumor efficacy and toxicity of BP-αCD3-αEGFR-ARC Exos.**
**A** Schematic of the animal study for BP-αCD3-αEGFR-ARC Exos in mice with BT-20 tumors. The left hindlimbs of NSG mice were inoculated with BT-20 cells via para-tibial injections ($n = 5$ or 7 per group). Expanded human PBMCs were intravenously injected on days 10 and 16 after cancer cells inoculation. One day post the first PBMCs injections, mice were administered (i.v.) with PBS, native exosomes (10 mg kg$^{-1}$), BP-ARC Exos (10 mg kg$^{-1}$), αCD3-αEGFR Exos (10 mg kg$^{-1}$), a combination of BP-ARC Exos (10 mg kg$^{-1}$) and αCD3-αEGFR Exos (10 mg kg$^{-1}$), CD9-CD38/αCD3-αEGFR Exos (10 mg kg$^{-1}$), or BP-αCD3-αEGFR-ARC Exos (10 mg kg$^{-1}$) every other day for six times. **B** Bioluminescence images of mice from all groups after the start of treatment. **C** Bioluminescence intensities of the treatment groups after the start of treatment ($n = 5$ or 7 per group; $^{***}p = 0.0009$ and $^{****}p < 0.0001$). Data are shown as mean ± SEM. Plasma levels of calcium (**D**; $^{***}p = 0.0004$) and TRAcP 5b (**E**; $^{***}p = 0.0002$) for each group of mice at the end of study ($n = 5$ or 7 per group). Data are presented as mean ± SD. **F** Bone mineral density (BMD) of left hindlimb bones for each group of mice at the end of study as quantified by micro-CT scanning ($n = 3$ per group; $^{**}p = 0.0030$). Data are shown as mean ± SD. **G** Representative micro-CT 3D images of left hindlimbs for each group of mice at the end of study. **H** Average body weights for each group of mice during the treatment study ($n = 5$ or 7 per group). Data are presented as mean ± SD. Plasma levels of creatinine (**I**) and ALT (**J**) for each group of mice at the end of study ($n = 5$ or 7 per group). Data are shown as mean ± SD. **K** Representative immunohistofluorescence images of tumor sections and quantitative analysis of tumor-infiltrating T cells for each group of mice at the end of study ($n = 40$ per group; $^{****}p < 0.0001$). Blue: nuclei stained with DAPI. Green: CD3$^+$ T cells stained with the anti-human CD3 antibody. Scale bars: 50 μm. Numbers of T cells were quantified with 20 fields of view per region and two mice per group. Data are shown in box-and-whisker plots with the line in the box as median and the lower and upper edge of the box as first and third quartile, respectively. Whiskers extending from the box represent the overall spread of the data. Statistical analysis was performed using two-way ANOVA with Dunnett's multiple comparison test for (**C**) and (**H**) or ordinary one-way ANOVA with Dunnett's multiple comparison test for (**D**–**F**) and (**I**–**K**). Significance of finding was defined as follows, ns = not significant $p > 0.05$, $^{**}p < 0.01$, $^{***}p < 0.001$, and $^{****}p < 0.0001$. Source data are provided as a Source Data file.

markers including calcium and tartrate-resistant acid phosphatase 5b (TRAcP 5b) at the end of study were determined. In comparison to other treatment groups, mice receiving BP-αCD3-αEGFR-ARC Exos were found with markedly lower concentrations of calcium and TRAcP 5b in plasma (Fig. 7D, E). Consistent with these results, micro-computed tomography (micro-CT) scanning of left hindlimbs revealed notably high levels of bone mineral density (BMD), bone volume, and bone surface for the BP-αCD3-αEGFR-ARC Exos-treated group relative to mice from other groups (Fig. 7F, G, and Supplementary Figs. 14 and 15). Also, reduced osteoclast numbers were detected in left hindlimbs of mice with BP-αCD3-αEGFR-ARC Exos relative to those from other groups (Supplementary Fig. 16). Moreover, mice body weights across the efficacy study, weight ratios of collected major organs, and structures of right tumor-free hindlimbs scanned by micro-CT suggest lack of noticeable toxicities (Fig. 7H, and Supplementary Fig. 17). No significant differences were seen among treatment groups for plasma levels of creatinine and alanine aminotransferase (ALT), a kidney and liver injury marker, respectively (Fig. 7I, J). These results together demonstrate superior efficacy and adequate safety of BP-αCD3-αEGFR-ARC Exos in eliminating tumors arising on bones.

Tumors were harvested at the end of animal study and subject to analysis by immunohistofluorescence and flow cytometry. While increased numbers of T cells were observed in tumors from mice receiving αCD3-αEGFR Exos, CD9-CD38/αCD3-αEGFR Exos, and the mixture (1:1) of BP-ARC Exos and αCD3-αEGFR Exos compared with those of mice injected with PBS, native exosomes, and BP-ARC Exos, BP-αCD3-αEGFR-ARC Exos-treated mice revealed the most profound T-cell tumor infiltration (Fig. 7K, and Supplementary Fig. 18). Moreover, flow cytometric analysis of T-cell subsets in spleen, blood, and tumor samples showed higher levels of CD8$^+$ T cells in tumors and CD8$^+$ and CD4$^+$ T cells in spleens for mice treated by exosomes expressing αCD3-αEGFR PDGFR TMD fusions than those of PBS-, native exosomes-, and BP-ARC Exos-treated animals (Supplementary Fig. 19). In contrast to other treatment groups, mice administered with BP-αCD3-αEGFR-ARC Exos were found with significantly enhanced populations of CD8$^+$ T cells in spleens and tumors (Supplementary Fig. 19). Collectively, these results suggest strong cellular immunity elicited by BP-αCD3-αEGFR-ARC Exos against tumors formed on bones.

## In vivo anti-metastasis efficacy and toxicity of bone-targeted ARC Exos specific for breast cancer and T cells

Considering that the bone-tumor niche can further promote the spread of cancer cells to other organs and tissues[58–62], we therefore assessed anti-metastatic activity of BP-αCD3-αEGFR-ARC Exos in mice bearing tumors derived from highly aggressive and invasive MDA-MB-231 cells. The left hindlimbs of NSG mice were implanted with luciferase-labeled MDA-MB-231 cells through para-tibial injections, followed by human PBMCs engraftments on days 7 and 13 (Fig. 8A). One day after the first PBMC injections, mice were intravenously given PBS, CD9-CD38/αCD3-αEGFR Exos, or BP-αCD3-αEGFR-ARC Exos every other day for six times. Whole body bioluminescence imaging indicated that both types of exosome treatments are efficacious in slowing tumor progression, but BP-αCD3-αEGFR-ARC Exos show enhanced anti-tumor activities compared with CD9-CD38/αCD3-αEGFR Exos (Supplementary Fig. 20). Consistent with the imaging results, histological analysis of left hindlimbs revealed limited sizes of tumors and retained bone mass and structure for animals with BP-αCD3-αEGFR-ARC Exos treatment over ones in other groups (Supplementary Fig. 21). Furthermore, BP-αCD3-αEGFR-ARC Exos-treated mice were characterized by declined levels of osteoclasts in left hindlimbs and bone resorption markers TRAcP 5b and calcium in plasma than other groups of mice (Supplementary Fig. 22).

Bioluminescence imaging of major organs and hindlimbs collected at the end of study revealed considerable metastatic cancer signals in kidney, lung, liver, brain, and right hindlimb for both PBS- and CD9-CD38/αCD3-αEGFR Exos-treated animals, but remarkably low extent of metastasis to these sites for mice receiving BP-αCD3-αEGFR-ARC Exos (Fig. 8B–D, and Supplementary Fig. 23). Unlike mice in PBS and CD9-CD38/αCD3-αEGFR Exos treatment groups which exhibited metastatic rates of 20–80% for kidney, 60% for brain, and 80–100% for lung, liver, and right leg, BP-αCD3-αEGFR-ARC Exos-treated mice showed no metastases in kidney, lung, brain, and right hindlimb and a metastatic rate of 16.7% in liver (Fig. 8E). Taken together, these results underscore superb efficacy of BP-αCD3-αEGFR-ARC Exos in suppressing growth and metastasis of tumors developing on bones.

In addition to anti-metastasis efficacy, toxicities were assessed for this mouse xenograft model of metastatic breast cancer. Body weight measurements during the study suggest no overt toxicities for exosome treatments (Supplementary Fig. 24a). Moreover, no major differences were seen for organ weight ratios and plasma concentrations of creatinine and ALT among three treatment groups (Supplementary Fig. 24b–d). Together with above toxicity assessments using the BT-20 tumor-based xenograft model, these data support great safety of BP-αCD3-αEGFR-ARC Exo.

Lastly, T cells were analyzed for collected tumor, spleen, and blood samples from all groups of animals. Immunohistofluorescence imaging of tumor sections revealed marked increases in tumor-infiltrating T cells for BP-αCD3-αEGFR-ARC Exos treatment group in comparison with PBS and CD9-CD38/αCD3-αEGFR Exos groups (Supplementary Fig. 25). Flow cytometric analysis showed elevated

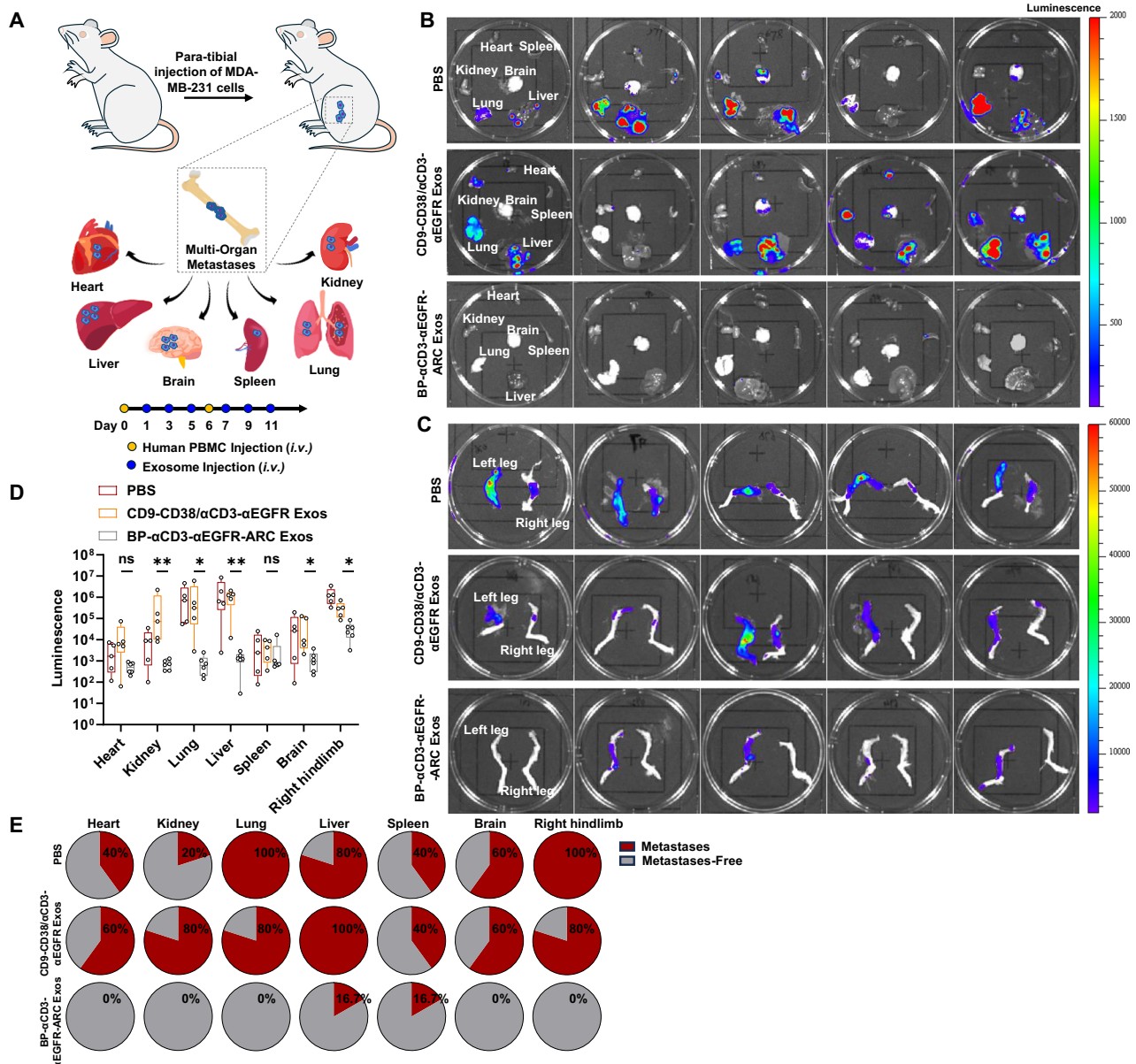

**Fig. 8 | In vivo anti-metastatic activity of BP-αCD3-αEGFR-ARC Exos.**
**A** Schematic of the anti-metastasis study for BP-αCD3-αEGFR-ARC Exos in mice with MDA-MB-231 tumors. The left hindlimbs of NSG mice were implanted with MDA-MB-231 cells via para-tibial injections study (*n* = 5 or 6 per group). Human PBMCs were intravenously injected on days 7 and 13 after tumor implantation. One day after the first PBMCs injections, mice were treated (i.v.) with PBS, CD9-CD38/αCD3-αEGFR Exos (10 mg kg⁻¹), or BP-αCD3-αEGFR-ARC Exos (10 mg kg⁻¹) every other day for six times. Representative luminescent images of major organs (**B**) and hindlimbs (**C**) for each group of mice at the end of study (*n* = 5 or 6 per group). **D** Luminescence intensities of major organs and right hindlimbs for each group of

mice at the end of study (*n* = 5 or 6 per group; kidney, ***p* = 0.0062; lung, *p* = 0.0151; liver, ***p* = 0.0096; brain, *p* = 0.0441; and right hindlimb, *p* = 0.0407). Data are shown in box-and-whisker plots with the line in the box as median and the lower and upper edge of the box as first and third quartile, respectively. Whiskers extending from the box represent the overall spread of the data. Each dot in the plot represents one biological replicate. Statistical analysis was performed using Kruskal–Wallis test followed by Dunn's multiple comparison test. Significance of finding was defined as follows, ns = not significant *p* > 0.05, *p* < 0.05, and ***p* < 0.01. **E** Metastatic rates to major organs and right hindlimbs for each group of mice at the end of study. Source data are provided as a Source Data file.

populations of CD8⁺ T cells in spleens and tumors of mice treated by CD9-CD38/αCD3-αEGFR Exos and BP-αCD3-αEGFR-ARC Exos (Supplementary Fig. 26a–c). Notably higher numbers of infiltrating CD8⁺ T cells were detected in tumors of mice treated by BP-αCD3-αEGFR-ARC Exos (Supplementary Fig. 26d). No significant differences in percentages of CD4⁺ T cells were found in spleen, blood, and tumor samples from these three groups (Supplementary Fig. 26e–g). Like above study using the mice with BT-20 tumors, these data support in vivo recruitment of cytotoxic T cells by BP-αCD3-αEGFR-ARC Exos toward attacking EGFR-expressing tumors on bones.

## Discussion
This work reports an approach of site-specific functionalization of exosome surface for new biological activities and/or specificities. The conjugation of functional groups of interest was achieved through utilizing surface-displayed catalytic domain of CD38 via a CD9 fusion protein and its derivatized covalent inhibitor 2′-Cl-araNAD⁺. By synthesizing 2′-Cl-araNAD⁺ conjugates carrying small-molecule cargos with distinct structures, chemically engineered exosomes could be facilely generated through one-step 2-h reactions in PBS at r.t. In addition to conventional fluorescent dyes, receptor ligands, cytotoxic

payloads, and bone-targeting agents could be efficiently attached to exosomes. The resulting Alexa 488-, FA-, MMAF-, FA-MMAF-, BP-, and BP-αCD3-αEGFR-ARC Exos demonstrated in vitro and/or in vivo functions and potential biomedical applications. In particular, BP-αCD3-αEGFR-ARC Exos, which recruit and activate cytotoxic T cells toward attacking EGFR-positive breast cancer cells as well as promote T-lymphocyte infiltration to tumors on bones, displayed enhanced therapeutic efficacy for tumor expansion and metastasis in different mouse xenograft models.

Cell-derived exosomes can be genetically modified for various purposes. However, this strategy is largely restricted to proteins and peptides expressed in exosomes. Additionally, genetic changes sometimes lead to unexpected issues with cargo sorting and enrichment, protein folding and stability, and proteolytic cleavage. In comparison, chemical approaches allow modifications to exosomes post-production. Moreover, exosomes can be armed with different classes of biomolecules beyond proteins. Established chemical methods exploit reactive groups incorporated into exosomes for random conjugations or hydrophobic tails for embedding functional groups into exosome membranes. The presented ARC Exo technology enables functionalization of exosomes with different types of small molecules in a site-specific manner, improving homogeneity and minimizing potential reduction or loss of activity and/or stability. In addition, the ARC Exo approach may allow to modify exosomes with other types of biomolecules such as nucleic acids and proteins difficult for genetic display. However, given their increased sizes, chemical modification sites and linker structures would need to be explored and optimized to ensure homogeneous and efficient conjugations without loss of functions.

To develop site-specifically functionalized exosomes, the enzymatic domain of CD38 was displayed on exosomal surface through genetic fusion with PDGFR TMD or CD9. Quantitative analysis of CD38 expression levels on exosomes by NGD+-based enzymatic activity assays, immunoblots, and ELISA showed greatly increased numbers of CD38 molecules expressed on exosomes with the CD9-CD38 fusion compared with those of CD38 detected on exosomes expressing the CD38-PDGFR fusion (Supplementary Fig. 1b–e). And fluorescence-based activity assays using NGD+ as a substrate indicated much higher catalytic activity for CD9-CD38 exosomes than that of CD38-PDGFR exosomes (Fig. 2C). Thus, the catalytic rate difference between CD9-CD38 exosomes and CD38-PDGFR exosomes likely results from differential expression levels of CD38 fusion proteins on exosomes rather than activity variations for two fusion designs. The enzymatic activities of CD38-PDGFR TMD and CD9-CD38 fusion proteins could also be different. The impact of TMD orientation on the fused CD38 activity can be analyzed using appropriate model systems. Compared with the single-pass PDGFR TMD, CD9 tetraspanin is characterized with high abundance in exosomes[63–65]. Fusion with CD9 could therefore lead to improved expression levels of CD38 on exosomes, giving rise to enhanced catalysis.

Exosome membranes with abundant proteins provide functional platforms for multivalent display of biomolecules with distinct activities and specificities. In contrast, it remains technically challenging to equip protein scaffolds such as immunoglobulins with multiple functional and/or targeting groups while retaining adequate stability and biological activity for each individual domain. Moreover, unlike multi-specific antibodies, the surfaced-engineered exosomes could be loaded with different forms of cargos for targeted delivery. It should be noted that in contrast to molecularly defined multi-functional antibodies, cell-derived exosomes are heterogeneous. Their compositions depend on origins and physiological states of parental cells and could alter physicochemical properties and biological functions. In addition to functioning as an ectoenzyme, CD38 is a known binding partner of several proteins such as CD31[66,67]. Surface-displayed CD38 may direct ARC Exos toward those cells expressing CD38-interacting proteins. Also, four mutated asparagine residues within the fused CD38 for elimination of glycosylation along with conjugated functional groups on ARC Exos may trigger host immune reactions. Therefore, future studies need to assess specificity, toxicity, and immunogenicity of ARC Exos.

In vivo studies showed significantly improved therapeutic efficacy against tumors established on bones for BP-αCD3-αEGFR-ARC Exos relative to CD9-CD38/αCD3-αEGFR Exos. Covalent attachments of BP molecules enable CD9-CD38/αCD3-αEGFR Exos to selectively bind to bones, especially for those EGFR-positive bone niches. In contrast to CD9-CD38/αCD3-αEGFR Exos, the bone-retained BP-αCD3-αEGFR-ARC Exos are likely to undergo reduced clearances, promoting recruitments of cytotoxic T cells and subsequent tumor infiltration. NSG mouse xenograft models with grafted human PBMCs facilitate in vivo assessments of anti-tumor immunity of BP-αCD3-αEGFR-ARC Exos targeting human cancer and T cells, but prevent evaluations of host immune responses and feasibility of repeated administration of BP-αCD3-αEGFR-ARC Exos, which are important for clinical translation. To comprehensively characterize immunoreactivities of this form of reprogrammed exosomes, immunocompetent mice along with mouse cell-derived ARC Exos would need to be used in the future.

In conclusion, ARC Exos carrying distinct types of small-molecule functional groups were generated via site-specific conjugations by combining surface-expressed CD38 enzyme with its covalent inhibitor 2′-Cl-araNAD+. With demonstrated in vitro and in vivo activities and specificities, ARC Exos offer tools for biomedical research and possibly a general approach for chemical engineering of exosomes at defined positions.

## Methods

### Materials
Ni-NTA resin (88221), xylenol orange tetrasodium salt (XO) (041379-04), and MTT 3-(4,5-dimethylthiazol- 2-yl)-2,5-diphenyltetrazolium bromide (M6494) were purchased from Thermo Fisher Scientific (MA, USA). PEI MAX transfection grade linear polyethyleneimine hydrochloride (24765-1) was ordered from Polysciences Inc. (PA, USA). BalanCD HEK293 medium (91165) was purchased from Irvine Scientific (CA, USA). Nicotinamide guanine dinucleotide (NGD+) (sc-215563) was from Santa Cruz Biotechnology (TX, USA). Multivette 600 LH-Gel (15.1675) for plasma preparation was ordered from SARSTEDT Group (NC, USA). Amicon ultra-15 centrifugal filter units with 10-kDa and 100-kDa molecular weight cutoff (UFC901008 and UFC910008), hydroxyapatite (HAp, 90023) and bovine serum albumin (BSA) (1265925GM) were purchased from Sigma-Aldrich (MO, USA). Dulbecco's modified Eagle's medium (DMEM) (10-017-CV) and RPMI 1640 medium (10-040-CV) were purchased from Corning Inc. (NY, USA).

### Cell culturing
HEK293T, BT-20, MDA-MB-231, and MDA-MB-453 cells from American Type Culture Collection (ATCC) and CAKI-1, SKOV-3, and A549 from Dr. Alan L. Epstein's laboratory at University of Southern California were cultured in DMEM medium (Corning, USA) supplemented with 10% FBS (VWR International, USA) at 37 °C and 5% CO2. Jurkat cells from ATCC and A2780 cells from Dr. Alan L. Epstein's laboratory at University of Southern California were cultured in RPMI 1640 medium supplemented with 10% FBS (VWR International, USA) at 37 °C and 5% CO2. Expi293F cells were purchased from Thermo Fisher Scientific and cultured with BalanCD HEK293 medium with shaking at 125 rpm and 37 °C in 5% CO2 incubators. Human PBMCs and T cells were purchased from Charles River Laboratories (MA, USA). Luciferase-expressing BT-20 and MDA-MB-231 cell lines were constructed using pCDH-luciferase, pMD2.G, and Pax2 plasmids.

## Molecular cloning and exosome expression

Synthetic gBLOCK DNA fragment encoding human full-length CD38 (Uniprot ID: P28907) with four mutated asparagine residues (N100D, N164A, N129D, and N209D) was ordered from Integrated DNA Technologies (IA, USA). To fuse CD38 with the N-terminus of human PDGFR TMD, the extracellular domain of CD38 (R45-I300) with a C-terminal GGGGS linker was amplified by polymerase chain reaction (PCR) and ligated into pDisplay vector (Thermo Fisher Scientific) between BglII and SalI restriction enzyme sites. The CD9-CD38 fusion protein was generated by fusing amplified full-length CD38 to the C-terminus of human CD9 through overlapping extension PCR. A flexible (GGGGS)$_2$ linker was inserted between CD9 and CD38 domains. The resulting DNA fragment encoding CD9-CD38 fusion was ligated into pDisplay vector between BglII and NotI restriction enzyme sites. Each CD38 fusion protein also contains an N-terminal hemagglutinin (HA) tag. A FLAG tag was included at C-terminus of the CD9-CD38 fusion. Primer used for PCR were ordered from Integrated DNA Technologies (IA, USA). Primer sequences are listed in Supplementary Table 2. The CD38 fusion constructions were confirmed by DNA sequencing. The plasmid expressing the αCD3-αEGFR-PDGFR fusion was constructed in a previous study[17].

To express fusion proteins in exosomes, Expi293F cells (240 mL) were transfected with the expression vectors for CD38-PDGFR (240 μg), CD9-CD38 (240 μg), αCD3-αEGFR-PDGFR (240 μg), or CD9-CD38 (80 μg) combined with αCD3-αEGFR-PDGFR (160 μg) at a density of $1.5 \times 10^6$ cells mL$^{-1}$ using PEI MAX 40K transfection reagent. After transfection, culture media containing exosomes were collected on days 3 and 6.

## Exosome purification

Exosomes were isolated through differential centrifugation and ultracentrifugation at 4 °C according to previous studies with modifications[17,18]. Briefly, cell cultures were first centrifuged at $100 \times g$ for 10 min to remove Expi293F cells. Then, supernatants were collected and centrifuged at $4000 \times g$ for 30 min to remove dead cells and at $14,000 \times g$ for 1 h to remove cell debris. Clarified supernatants were next centrifuged at $256,000 \times g$ for 1.5 h with a Type 70 Ti rotor by Optima L-80 XP ultracentrifuge (Beckman Instruments) to pellet exosomes. After washing with PBS twice, pellets were resuspended in PBS and filtered with 0.22 μm membranes. Protein concentrations of exosomes were determined by Bradford protein assay kit by following manufacturer's instructions.

## Expression and purification of recombinant CD38 extracellular domain

The plasmid expressing the extracellular domain of human CD38 (R45-I300) with a C-terminal His$_6$-tag was constructed in a previous study[38]. Briefly, CD38 extracellular domain was expressed in Expi293F cells through transient transfection. On day 5 following transfection, media containing the secreted CD38 extracellular domain were collected and loaded onto Ni-NTA gravity columns. After extensive washing with a wash buffer (20 mM Tris-HCl, pH 8.0, 200 mM NaCl, 30 mM imidazole), CD38 extracellular domain was eluted by an elution buffer (20 mM Tris-HCl, pH 8.0, 200 mM NaCl, 400 mM imidazole). After dialysis in 20 mM Tris-HCl (pH 8.0) with 300 mM NaCl, purified CD38 extracellular domain was concentrated with 10-kDa molecular weight cut-off concentrators.

## Immunoblot analysis

Exosomes (5 μg of protein) were boiled and separated in 4–20% Express Plus-PAGE gels (GeneScript, Piscataway, NJ, USA), followed by transferring to immun-blot PVDF (polyvinylidene fluoride) membranes (Bio-Rad Laboratories, CA, USA). After blocking with 5% BSA in PBST (PBS with 0.1% Tween-20) for 1 h at room temperature, membranes were incubated with anti-HA (2-2.2.14, Thermo Fisher Scientific, diluted 1:1000 in PBST with 5% BSA), anti-CD38 (HIT2, BioLegend, CA, USA, diluted 1:500 in PBST with 5% BSA), anti-CD9 (D8O1A, Cell Signaling Technology, MA, USA, diluted 1: 1:1000 in PBST with 5% BSA), anti-CD63 (H5C6, BioLegend, CA, USA, diluted 1:1000 in PBST with 5% BSA), and anti-CD81 (1.3.3.22, Thermo Fisher Scientific, diluted 1:1000 in PBST with 5% BSA) for 4 h at room temperature. Following incubation with appropriate secondary IgG antibody-HRP conjugates (Thermo Fisher Scientific, diluted 1:3000 in PBST) for 2 h at room temperature, membranes were developed and imaged using a ChemiDoc Touch Imaging System (Bio-Rad Laboratories).

## Nanoparticle tracking analysis

The size distribution and concentration of exosomes were measured via NTA using a Nanosight LM10 (Malvern Instruments, UK) by following the manufacturer's instructions[19,22,23]. Each sample was analyzed for 60 s with six replicates.

## Enzymatic activity of exosomes

Exosomes (10 μg mL$^{-1}$) and CD38 protein (20 nM) were incubated with 100 μM of NGD$^+$ in PBS. Using a Synergy H1 plate reader, the formation of cyclic guanine dinucleotide phosphate-ribose (cGDPR) was monitored for 10 min based on fluorescence intensity at 410 nm emission and 300 nm excitation.

To examine the enzymatic activity by HPLC, NAD$^+$ (200 μM) was incubated with native exosomes (50 μg mL$^{-1}$), CD9-CD38 exosomes (50 μg mL$^{-1}$), or CD38 protein (200 nM) in PBS for 4 h at room temperature. The reaction mixtures were analyzed at 260 nm UV absorbance using a semipreparative C18 Kinetex column (5 μm, 100 Å, 150 × 4.6 mm, Phenomenex Inc, Torrance, CA) with following condition (mobile phase A: 0.1% formic acid in water; mobile phase B: 0.1% formic acid in acetonitrile; flow rate = 1.0 mL min$^{-1}$; 0–10 min: 0–5% B, 10–12 min: 5–0% B).

## Chemical synthesis of 2′-Cl-araNAD$^+$-derived conjugates

Experimental details are included in Supplementary Method 1 for the synthesis of 2′-Cl-araNAD$^+$-folic acid (FA) (Supplementary Figs. 27–43), 2′-Cl-araNAD$^+$-bisphosphonate (BP) (Supplementary Figs. 44–49), and 2′-Cl-araNAD$^+$-monomethyl auristatin F (MMAF) (Supplementary Fig. 50).

## Exosome conjugation

To generate exosomes conjugated with Alexa Fluor 488, the linker 2′-Cl-araNAD$^+$-N$_3$ (10 μM) was first incubated with exosomes (0.1 mg mL$^{-1}$) at room temperature for 2 h. Following buffer exchange with PBS using 100-kDa molecular weight cut-off filters, exosomes with conjugated linkers were then incubated with Alexa Fluor 488-DBCO (10 μM) in PBS for 60 min on ice. Free Alex Fluor 488-DBCO was removed by buffer exchange with PBS.

To make exosomes conjugated with FA, MMAF, and BP, purified exosomes (0.1 mg mL$^{-1}$) were incubated with the synthesized 2′-Cl-araNAD$^+$-FA (10 μM), 2′-Cl-araNAD$^+$-MMAF (10 μM), and 2′-Cl-araNAD$^+$-BP (10 μM) in PBS at room temperature for 2 h. Conjugation was monitored by NGD$^+$-based fluorescence assays. Upon completion, reactions were exchanged to PBS with 100-kDa molecular weight cut-off filters to remove free 2′-Cl-araNAD$^+$ conjugates.

## CD38 conjugation

To prepare recombinant CD38 protein conjugated with MMAF, purified CD38 extracellular domain (1 μM) were incubated with the synthesized 2′-Cl-araNAD$^+$-MMAF (20 μM) in PBS buffer at room temperature for 2 h. Conjugation was monitored by NGD$^+$-based fluorescence assays. Free 2′-Cl-araNAD$^+$-MMAF was removed with 10-kDa molecular weight cut-off concentrators upon completion of conjugation.

## Quantification of expression levels of CD38 on exosomes

The expression levels of CD38 molecules on exosomes were quantified by NGD$^+$-based enzymatic activity assays, immunoblots, and ELISA assays. Recombinant CD38 was purified and used as standards.

For measuring CD38 enzymatic activity, CD38-expressing exosomes and purified recombinant CD38 were incubated with $100\,\mu M$ NGD$^+$ in PBS. Fluorescence signals at 410 nm emission and 300 nm excitation were recorded. The expression levels of CD38 on exosomes were determined based on standard curves established from reaction rates of recombinant CD38 at varied concentrations.

For immunoblot analysis, CD38-expressing exosomes and recombinant CD38 were detected with an anti-CD38 primary antibody (HIT2, BioLegend, CA, USA) and a goat anti-mouse-HRP secondary antibody. The CD38 band intensities were quantified using Fiji software (Image J). The expression levels of CD38 on exosomes were calculated by referring to standard curves created from recombinant CD38 protein.

For the ELISA method, CD38-expressing exosomes and recombinant CD38 were coated onto 96-well ELISA plates for 8 h at 4 °C. After washing with $300\,\mu L$ of PBS buffer four times, plates were blocked with 3% BSA in $300\,\mu L$ of PBS for 2 h at room temperature. Following 2-h incubation with an anti-CD38 antibody ($2\,\mu g\,mL^{-1}$, HIT2, BioLegend, CA, USA) in $100\,\mu L$ of PBS containing 0.3% BSA at room temperature, wells were washed with $300\,\mu L$ of PBS buffer four times and then incubated with a goat anti-mouse secondary antibody-HRP for 30 min. After washing with PBS buffer, $100\,\mu L$ of QuantaBlu fluorogenic substrate was added. Fluorescence intensities were measured at 420 nm emission and 325 nm excitation using a Synergy H1 plate reader. The expression levels of CD38 on exosomes were derived according to recombinant CD38 standard curves.

## Nanoflow cytometry

Non-stained CD9-CD38 Exos, Alexa 488-ARC Exos, and PKH67 labeled CD9-CD38 Exos were analyzed using a Cytek Aurora flow cytometer (Fremont, CA, USA). Side scatter (SSC) and fluorescence intensity of exosomes ($1\times 10^5$) were measured and analyzed using FlowJo-V10 software (Tree Star, USA).

## Binding to recombinant human FRα by ELISA

Various forms of exosomes at final concentrations of 0.02, 0.06, 0.2, 0.6, 1.9, 5.6, 16.7, and $50\,\mu g\,mL^{-1}$ were coated onto 96-well ELISA plates overnight at 4 °C. After rinsing with $400\,\mu L$ of PBST (PBS with 0.05% Tween-20) three times, wells were blocked with 3% BSA in $300\,\mu L$ of PBS for 2 h at room temperature. Recombinant human FRα protein (11241-H08H, Sino Biological, TX, USA) at a final concentration of $1\,\mu g\,mL^{-1}$ in $100\,\mu L$ of PBS containing 0.5% BSA was added and incubated for 2 h at room temperature. After washing with $400\,\mu L$ of PBST, $100\,\mu L$ of anti-His$_6$ antibody (HIS.H8, Thermo Fisher Scientific, $0.5\,\mu g\,mL^{-1}$) was added and incubated for 60 min at room temperature. Following 30-min incubation with goat anti-mouse secondary antibody-HRP and three washes with PBST, $100\,\mu L$ of QuantaBlu fluorogenic substrate was added into the wells. Using a Synergy H1 plate reader, fluorescence intensities were measured at 420 nm emission and 325 nm excitation. The EC$_{50}$ values were analyzed by Graph-Pad Prism (GraphPad Software, CA, USA) using a sigmoidal dose-response model.

## Flow cytometry of antigen expression

To analyze cell-surface expression levels of FOLR1 for CAKI-1, A2780, SKOV-3, and A549 cell lines, cells were stained with PE anti-FOLR1 antibody (LK26, BioLegend, diluted 1:100 in DPBS with 1% FBS) on ice. To analyze cell-surface expression levels of EGFR or CD3 for MDA-MB-231, BT-20, MDA-MB-453, and Jurkat cell lines, cells were stained with Alexa Fluor 488 anti-EGFR antibody (AY13, BioLegend, diluted 1:100 in DPBS with 1% FBS) or FITC anti-CD3 antibody (UCHT1, BioLegend,

diluted 1:100 in DPBS with 1% FBS) on ice. After three washes with PBS, cells were analyzed using a BD Fortessa X20 flow cytometer (BD Biosciences, CA). All flow cytometry data were processed by FlowJo-V10 software (Tree Star, USA).

## HAp binding assay

BP-modified exosomes or non-modified exosomes were diluted in 0.5 mL of PBS. HAp (20 equiv, 10 mg) was added, and then the mixture was shaken at 220 rpm and 37 °C. Samples without HAp were used as controls. After 0.25, 0.5, 1, 2, 4, 8, and 24 h, the suspensions were centrifuged ($6000\times g$, 1 min), and the concentrations of the supernatants were measured with Pierce Bradford plus protein assay reagent. The percent binding to HAp was calculated as follows:

$$\text{HAp binding percentage} = \left(1 - C_{\text{with HAp}}/C_{\text{without HAp}}\right)\times 100\% \quad (1)$$

where C represents the exosome concentration.

## Preparation of non-decalcified bone sections

Fresh mouse hindlimb bones were fixed in 10% buffered formalin at 4 °C for 72 h, followed by resining with running tap water for 2 h and subsequent dehydration with 70%, 95%, 100% ethanol for 4 day. Bones were infiltrated first using 100% methyl methacrylate (MMA) for 4 day and then MMA II buffer (21 g dibenzoyl peroxide (BPO) per liter of MMA) for 4 day with two buffer changes. Bone samples were next incubated with MMA III buffer (210 g BPO and 250 mL dibutyl phthalate per liter of MMA) in a 30 °C water bath for material polymerization for 10 day. Bone sections were prepared using a microtome (Microm HM 3100).

## Native bone binding assay

Native bones of mice were cut into small fragments, followed by washing with PBS and anhydrous ethanol and drying at room temperature overnight. To examine binding to bones, 0.5 mg BP-modified exosomes or non-modified exosomes was mixed with native bone fragments (30 mg) in 0.5 mL of PBS. Then, the mixtures were shaken at 220 rpm and 37 °C. Samples without bone fragments were used as controls. The suspensions were centrifuged ($5000\times g$, 1 min) after 0.25, 0.5, 1.0, 2.0, 4.0, 6.0, and 12 h, and the concentrations of the supernatants were measured with Pierce Bradford plus protein assay reagent. The percent binding to native bone was calculated according to the following equation:

$$\text{Bone binding percentage} = (1 - C_{\text{with bone}}/C_{\text{without bone}})\times 100\% \quad (2)$$

where C represents the exosome concentration.

## Binding of exosomes to target cells by flow cytometry

Jurkat, MDA-MB-231, BT-20, and MDA-MB-453 cells were incubated with exosomes ($0.1\,mg\,mL^{-1}$) in PBS for 30 min on ice. After washing three times with PBS, cells were incubated with anti-HA antibody (2-2.2.14, Thermo Fisher Scientific, diluted 1:200 in DPBS with 1% FBS) for 60 min on ice. Following three washes with PBS, cells were incubated with an Alexa Fluor 488 anti-mouse IgG (H+L) secondary antibody (A-11001, Thermo Fisher Scientific, diluted 1:200 in DPBS with 1% FBS) for 30 min on ice. Samples were washed three times with PBS and analyzed using a BD Fortessa X20 flow cytometer. All flow cytometry data were processed with FlowJo-V10 software.

## Labeling of exosomes with PKH67

Exosomes were labeled with PKH67 dye (MINI67, Sigma-Aldrich) by following the manufacturer's instructions. Briefly, exosomes ($0.2\,mg\,mL^{-1}$) were diluted and incubated with PKH67 dye ($1\,\mu M$) in PBS for 30 min at 37 °C. Following buffer exchange with PBS, labeled exosomes were concentrated with 100-kDa molecular weight cut-off concentrators.

## Confocal microscopy

To examine cellular uptake of exosomes, HEK293T cells were seeded around 60% confluency and incubated overnight. Alexa 488-ARC Exos and other exosome control groups (50 μg mL$^{-1}$) were incubated with HEK293T cells for 2 h on ice or at 37 °C. After washing with PBS, cells were fixed with by 4% paraformaldehyde (PFA) for 30 min at 4 °C and then stained with Hoechst 33342 (1 μg mL$^{-1}$) (H1399, Thermo Fisher Scientific) for 15 min at room temperature. Following three washes with PBS, cells were imaged by a Lecia SP8 microscope (Leica Microsystems, IL, USA).

To visualize the binding of FA-ARC Exos to FRα-positive cells, CAKI-1, A2780, SKOV-3, and A549 cells were grown to ~60% confluency in confocal dishes. The cells were incubated with 50 μg mL$^{-1}$ PKH67-labeled exosomes for 30 min at 4 °C and then fixed by 4% PFA for 30 min at 4 °C. The cells were washed three times with PBS containing 1 μg mL$^{-1}$ Hoechst 33342 for 15 min and then washed three times with PBS.

To quantify the uptake efficiency of MMAF-ARC Exos and FA-MMAF-ARC Exos, MMAF-ARC Exos and FA-MMAF-ARC Exos were labeled with PKH67. CAKI-1, A2780, SKOV-3, and A549 cells were incubated with PKH67-labeled exosomes for 4 h at 37 °C. After fixation with 4% PFA for 30 min at 4 °C, cells were stained with 1 μg mL$^{-1}$ Hoechst 33342 and washed with PBS.

To assess the binding of BP-αCD3-αEGFR-ARC Exos to T cells, BT-20 cells, and bone tissues, non-decalcified bone sections were incubated with T cells and BT-20 cells in confocal dishes. Native exosomes, CD9-CD38/αCD3-αEGFR Exos, and BP-αCD3-αEGFR-ARC Exos were labeled with PKH67 and added into confocal dishes for 30-min incubation at 4 °C. Samples were then gently washed with PBS, fixed with 4% PFA for 30 min at 4 °C, and stained with 1 μg mL$^{-1}$ Hoechst 33342.

Confocal microscopic images of cells were collected using a Lecia SP8 microscope. Fluorescence intensities of images were quantified using Fiji software (Image J).

## Quantification of MMAF levels on MMAF-ARC Exos and FA-MMAF-ARC Exos

MMAF-conjugated CD38 standards were prepared by incubating 20 μM 2′-Cl-araNAD$^+$-MMAF with 1 μM recombinant CD38 extracellular domain in PBS buffer for 2 h at room temperature. Free 2′-Cl-araNAD$^+$-MMAF was removed using 10-kDa molecular weight cut-off concentrators. Exosomes and MMAF-CD38 were coated on 96-well ELISA plates overnight at 4 °C. After washing with 300 μL of PBS buffer four times, plates were blocked with 3% BSA in 300 μL of PBS for 2 h at room temperature. Following incubation with an anti-MMAF antibody (2 μg mL$^{-1}$, MME-M5252, ACROBiosystems, DE, USA) in 100 μL of PBS containing 0.3% BSA for 2 h at room temperature, plates were washed with PBS and incubated with a goat anti-mouse secondary antibody-HRP (diluted 1:2000 in PBST) for 30 min. After rinsing with PBS, QuantaBlu fluorogenic substrate was added. Fluorescence signals at 420 nm emission and 325 nm excitation were measured. The levels of MMAF on MMAF-ARC Exos and FA-MMAF-ARC Exos were calculated based on standard curves of MMAF-CD38.

## Stability of MMAF-ARC Exos in mouse plasma

MMAF-ARC Exos (0.1 mg mL$^{-1}$) were incubated with fresh mouse plasma at 37 °C for 0, 4, 12, 24, 48, 96, 168, 216, 264 and 336 h. Samples were analyzed by immunoblot with an anti-MMAF antibody (MME-M5252, ACROBiosystems, DE, USA, diluted 1:500 in DPBS). Intact band intensities were quantified using Fiji software (Image J). The half-life value was calculated using GraphPad Prism.

## Flow cytometry of binding of BP-αCD3-αEGFR-ARC Exos to BT-20 and T cells

T cells (2 × 10$^5$ cells per well) and BT-20 cells (2.5 × 10$^4$ cells per well) were mixed in RPMI 1640 medium with 10% FBS and incubated in 24-well plates. PKH67-labeled BP-αCD3-αEGFR-ARC Exos were added into

wells and incubated at 37 °C for 0, 15, 30, 60, 120, and 240 min. After washing with PBS, cells were analyzed using a BD Fortessa X20 flow cytometer. Flow cytometry data were processed by FlowJo-V10 software (Tree Star, USA).

## Binding to bone sections

Non-decalcified bone sections from mice were blocked with 3% BSA for 1 h at room temperature. After washing with PBS, PKH67-labeled exosomes (50 μg mL$^{-1}$) were added and incubated for 1 h at 37 °C After washing three times with PBS, specimens were incubated for 30 min at 37 °C with XO (2 μg mL$^{-1}$) in PBS. After four washes with PBS, slides were then sealed with Fluoromount-G mounting medium (F4680-25ML, Sigma-Aldrich). Confocal microscopic images were collected using a Lecia SP8 microscope.

## In vitro cytotoxicity

To analyze the cytotoxicity of exosomes for CAKI-1, A2780, SKOV-3 and A549 cell lines, cells were seeded in the 96-cell plates with each well containing 5000 cells overnight. After treatment with various concentration of exosomes at 37 °C with 5% CO$_2$ for 42 h, MTT reagent was added and incubated for 2 h at 37 °C with 5% CO$_2$. Next, lysis buffer (10% SDS-HCl solution, 100 μL) was added, followed by 2-h incubation at room temperature. Absorbance was measured at 580 nm using a Synergy H1 plate reader. Cells treated with 5 μM paclitaxel and culture medium were included as blank and negative controls, respectively.

To measure the cytotoxicity of CD9-CD38 Exos for human PBMCs, non-activated PBMCs and PBMCs activated with an anti-human CD3 antibody (10 μg mL$^{-1}$, OKT3, BioLegend) and an anti-human CD28 antibody (2.5 μg mL$^{-1}$, CD28.2, BioLegend) were seeded in 96-well plates (5 × 10$^4$ cells per well) and treated with CD9-CD38 Exos for 48 h at 37 °C with 5% CO$_2$. MTT reagent was added and incubated for 2 h at 37 °C with 5% CO$_2$. After additions of lysis buffer (10% SDS-HCl solution, 100 μL) and incubation for 2 h at room temperature, absorbances at 580 nm were recorded using a Synergy H1 plate reader.

To evaluate the cytotoxicity of exosomes for MDA-MB-231, BT-20, and MDA-MB-453 cells (target cells) in the presence of human PBMCs (effector cells), target cells (5 × 10$^3$ cells per well) were mixed with non-activated human PBMCs (4 × 10$^4$ cells per well) and incubated with various concentrations of exosomes in 96-well plates for 48 h at 37 °C with 5% CO$_2$. After washing with PBS and removals of human PBMC suspensions, wells were added with MTT reagent and incubated for 2 h at 37 °C with 5% CO$_2$. Lysis buffer (10% SDS-HCl solution, 100 μL) was then added and incubated at room temperature for 2 h. Absorbance at 570 nm was measured using a Synergy H1 plate reader. The EC$_{50}$ values were determined using GraphPad Prism.

## T-cell activation analysis

T cell activation assays were performed according to previous studies with modifications[18,22]. Briefly, non-activated human PBMCs (2 × 10$^5$ cells per well) were incubated with target cells (2.5 × 10$^4$ cells per well) in the presence of PBS or exosomes at 1 μg mL$^{-1}$ in 24-well plates for 48 h. Cells were stained with Alexa Fluor 488 anti-human CD3 (UCHT1, BioLegend, diluted 1:100 in DPBS with 1% FBS), PE anti-human CD25 (M-A251, BioLegend, diluted 1:100 in DPBS with 1% FBS), and APC anti-human CD69 (FN50, BioLegend, diluted 1:200 in DPBS with 1% FBS) on ice for 30 min, followed by flow cytometric analysis using a BD Fortessa X20 flow cytometer. The concentrations of secreted interferon gamma (IFN-γ) and interleukin-2 (IL-2) in collected culture media were measured using ELISA kit (DY285B and DY202, R&D System, MN, USA).

To measure effects of CD9-CD38 Exos on functions of human PBMCs, non-activated PBMCs and PBMCs activated with an anti-human CD3 antibody (10 μg mL$^{-1}$, OKT3, BioLegend) and an anti-human CD28 antibody (2.5 μg mL$^{-1}$, CD28.2, BioLegend) were incubated in 24-well plates (2 × 10$^5$ cells per well) and treated with CD9-CD38 Exos (20, 100, 500, 2000, and 10,000 ng mL$^{-1}$) for 48 h at 37 °C

with 5% $CO_2$. The media were collected after centrifugation at $500 \times g$ for 10 min. The levels of secreted IFN-γ and IL-2 in culture media were measured using ELISA kit mentioned above.

## Biodistribution

Six-week female NOD.Cg-*Prkdc*$^{scid}$ *Il2rg*$^{tm1Wjl}$/SzJ (NSG) mice were ordered from the Jackson Laboratory (Bar Harbor, ME, USA). All animal experiments were approved by the Institutional Animal Care and Use Committee of the University of Southern California. Mice were provided with normal food and water in a pathogen-free environment with a 12-h light/12-h dark cycle, temperature of ~25 °C, and humidity of ~50%. No tumors in the experiments exceeded the maximal size (1800 mm$^3$). Mice were inoculated with BT-20 cells ($2 \times 10^5$ cells per mouse) via para-tibial injections into left hindlimbs. After 3 weeks, DiR-labeled exosomes (10 mg kg$^{-1}$) were administered to tumor-bearing mice by tail vein injections. Fluorescence signals of hindlimbs and teeth at 24, 48, 96, and 168 h after exosome injections were measured with an IVIS Lumina III imaging system (PerkinElmer, MA, USA). The biodistribution ratio of BP-αCD3-αEGFR-ARC Exos for tumor bone tissue to normal bone tissue at each time point was calculated based on fluorescence intensities.

## In vivo anti-tumor efficacy of BP-αCD3-αEGFR-ARC Exos

Six-week female NSG mice were inoculated with luciferase-expressing BT-20 cells ($2 \times 10^5$ cells per mouse) via para-tibial injections into left hindlimbs. All animal experiments were approved by the Institutional Animal Care and Use Committee of the University of Southern California. Mice were provided with normal food and water in a pathogen-free environment with a 12-h light/12-h dark cycle, temperature of ~25 °C, and humidity of ~50%. No tumors in the experiments exceeded the maximal size (1800 mm$^3$). Human PBMC ($2 \times 10^6$ cells mL$^{-1}$) were cultured in RPMI 1640 medium with 10% FBS and stimulated with immobilized anti-human CD3 antibody (10 µg mL$^{-1}$, OKT3, BioLegend) and soluble anti-CD28 antibody (2 µg mL$^{-1}$, 28.2, BioLegend) in flasks for 3 days. Cells were then expanded with human IL-2 (40 IU mL$^{-1}$, BioLegend) in RPMI 1640 medium plus 10% FBS for 4–6 days. Mice were administered with the expanded human PBMCs ($2 \times 10^7$ cells per mouse) via tail vein intravenous injections on days 10 and 16 after tumor cells inoculation. One day after the first human PBMC injections, mice were randomized into seven groups and intravenously treated via tail vein injections with PBS, native Exos (10 mg kg$^{-1}$), BP-ARC Exos (10 mg kg$^{-1}$), αCD3-αEGFR exosomes (10 mg kg$^{-1}$), a combination of BP-CD38 Exos (10 mg kg$^{-1}$) and αCD3-αEGFR exosomes(10 mg kg$^{-1}$), CD9-CD38/αCD3-αEGFR exosomes (10 mg kg$^{-1}$), and BP-αCD3-αEGFR-ARC exosomes (10 mg kg$^{-1}$) every other day for a total of six times. The animals were imaged once a week using an IVIS Lumina III imaging system after intraperitoneal injection of D-luciferin (150 mg kg$^{-1}$) (Syd Labs, MA, USA). At the end of the study, mice were euthanized, and major organs and tissues were collected for weighing and further analysis.

Blood, spleen, and tumor tissues from all groups were prepared for single-cell suspensions. Cells were treated with ACK lysis buffer (A1049201, Thermo Fisher Scientific) by following the manufacturer's instructions. After staining for live and dead cells with live/dead-fixable Zombie Aqua (423101, BioLegend, 1 µM), cells were washed three times and stained with PerCP/Cyanine5.5 anti-human CD45 antibody (2D1, BioLegend, diluted 1:100 in DPBS with 1% FBS), FITC anti-human CD3 antibody (UCHT1, BioLegend, diluted 1:100 in DPBS with 1% FBS), APC/Cyanine7 anti-human CD4 antibody (OKT4, BioLegend, diluted 1:50 in DPBS with 1% FBS), and Pacific Blue anti-human CD8 antibody (RPA-T8, BioLegend, diluted 1:100 in DPBS with 1% FBS). Following three washes, 50,000 cells were analyzed by a BD Fortessa X-20 Cell Analyzer. All flow cytometry data were processed with FlowJo-V10 software. Data for all samples were first gated based on morphology (FSC-A/SSC-A) to remove non-cell debris. Zombie Aqua-staining negative (live cells) populations were then analyzed on fluorescence intensities of different antibodies used for the staining.

## Radiographic analysis

Left and right hindlimbs of mice from all treatment groups were collected at the end of above animal efficacy study and scanned by microcomputed tomography (micro-CT, Scanco Medical µCT 50, SCANCO Medical AG, Switzerland) at a resolution of 6.5 µm/pixel. Raw images were reconstructed and analyzed using a region of interest in Scanco Medical Evaluation software 1.1.19.0.

## In vivo anti-metastasis efficacy of BP-αCD3-αEGFR-ARC Exos

All animal experiments were approved by the Institutional Animal Care and Use Committee of the University of Southern California. Mice were provided with normal food and water in a pathogen-free environment with a 12-h light/12-h dark cycle, temperature of ~25 °C, and humidity of ~50%. No tumors in the experiments exceeded the maximal size (1800 mm$^3$). Six-week female NSG mice were inoculated with luciferase-expressing MDA-MB-231 cells ($5 \times 10^5$ cells per mouse) via para-tibial injections into left hindlimbs. Human PBMCs were expanded as described above. Mice were administered with the expanded human PBMCs ($2 \times 10^7$ cells per mouse) via tail vein intravenous injections on days 7 and 13 after cancer cells inoculation. One day post human PBMCs injections, mice were randomized into three groups and intravenously treated via tail vein injections with PBS, CD9-CD38/αCD3-αEGFR exosomes (10 mg kg$^{-1}$), and BP-αCD3-αEGFR-ARC exosomes (10 mg kg$^{-1}$) every other day for a total of six times. The animals were imaged twice a week using an IVIS Lumina III imaging system after intraperitoneal injection of D-luciferin (150 mg kg$^{-1}$). At the end of the study, mice were euthanized, and blood, hearts, livers, spleens, lungs, kidneys, brains, and tibia bones were collected. Bioluminescence imaging of the collected organs and tissues was immediately performed using an IVIS Lumina III imaging system. Blood, spleens, and tumors from all groups were prepared for single-cell suspensions and analyzed by flow cytometry as described above.

## Plasma levels of calcium, TRAcP 5b, creatinine, and ALT

At the end of animal studies, blood samples for all groups were collected with Multivette 600 LH-Gel and centrifuged for 10 min at $5000 \times g$ to obtain plasma. The calcium levels in plasma were determined by using calcium detection kit (DICA-500, Bioassays, Hayward, CA, USA). The concentrations of osteoclast-derived TRAcP 5b in plasma were measured by mouse TRAcP-5b ELISA kit (orb1146716, Biorbyt, Durham, NC, USA).

To quantify the concentrations of creatinine in plasma, plasma samples (20 µL) were mixed with equal volume of trichloroacetic acid (1.2 M, Oakwood Chemicals, SC, USA). After centrifugation at $12,000 \times g$ for 20 min, supernatants (30 µL) were incubated with working solution (60 µL, 38 mM picric acid premixed with equal volume of 1.2 M NaOH). Creatinine standards at varied concentrations were also prepared for generating standard curves on the same plates. After incubation at room temperature for 20 min, absorbance at 510 nm was determined by a Synergy H1 plate reader.

To measure ALT activities in plasma, plasma samples (5 µL) or standard sodium pyruvate (5 µL) was first incubated for 1 h at 37 °C with substrate solution (25 µL, 200 mM alanine and 2 mM 2-oxoglutarate in 100 mM disodium phosphate, pH 7.4), followed by additions of 2,4-dinitrophenylhydrazine (25 µL, 1 mM in 1 M HCl) and 20-min incubation at room temperature. Reactions were then mixed with 0.5 M NaOH (250 µL) and measured for absorbance at 510 nm using a Synergy H1 plate reader.

## Immunofluorescence and histology analysis

Immunofluorescence staining of tumors collected at the end of animal studies was performed on 6 µm cryosections. The tissues were fixed with 4% PFA for 10 min at room temperature and blocked with PBS containing 3% BSA for 1 h at room temperature. The tissue sections were then incubated with anti-human CD3 antibody (UCHT1,

BioLegend, 1 μg mL$^{-1}$) overnight at 4 °C, followed by washing with PBS. The sections were then incubated with Alexa Fluor 488 anti-mouse IgG (H+L) secondary antibody (diluted 1:200 in DPBS) for 1 h at room temperature and next stained with Hoechst 33342. Images were acquired using a Leica SP8 microscope. Twenty random non-overlapping regions of each tumor ($n = 2$ mice/group) were imaged for quantification.

Harvested tibias from each group at the end of animal studies were fixed in 4% PFA for 2 d and then decalcified in 12% EDTA at 4 °C for 2 weeks. After washing with PBS, samples were embedded in paraffin by following standard procedures. Five μm sections were prepared on glass slides. After drying in an oven and then deparaffinization with xylene solution, hematoxylin and eosin (H&E) staining were performed via the conventional method. Masson-Goldner trichrome staining and tartrate-resistant acid phosphatase (TRAP) staining were performed using Masson-Goldner staining kit (1.00485, Sigma-Aldrich) and TRAP staining kit (387 A, Sigma-Aldrich), respectively. Images were collected with Cytation 5 (BioTek, VT, USA).

### Statistical analysis
The statistical analyses were performed using GraphPad Prism on independent biological replicates, $n \geq 3$ for in vitro assays and $n \geq 5$ for in vivo studies. Statistical significance between two groups was determined using two-tailed student $t$ tests. One- or two-way ANOVA analyses were performed for comparisons of multiple groups. Significance of finding was defined as follows: ns = not significant $p > 0.05$, $^{*}p < 0.05$, $^{**}p < 0.01$, $^{***}p < 0.001$, and $^{****}p < 0.0001$.

### Reporting summary
Further information on research design is available in the Nature Portfolio Reporting Summary linked to this article.

## Data availability
Data supporting the findings of this work are available within the paper and its Supplementary Information files. A reporting summary for this Article is available as a Supplementary Information file. Source data are provided with this paper.

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

## Acknowledgements

The authors acknowledge technical support for nanoflow cytometry by the Flow Cytometry and Immune Monitoring core of University of Southern California supported by National Institutes of Health (NIH) grant P30CA014089. This work was supported in part by National Institute of General Medical Sciences (NIGMS) of NIH grant R35GM137901 (to Y. Z.), National Institute of Biomedical Imaging and Bioengineering (NIBIB) of NIH grant R01EB031830 (to Y. Z.), and National Cancer Institute (NCI) of NIH grant R01CA276240 (to Y. Z.).

## Author contributions

L.Z. and Y.Z. designed research. L.Z., S.S., A.J.A., G.K., S.H.K., Z.Z., K.S., T.H., X.D., Q.C., and T.S. performed research. L.Z., S.S., A.J.A., and Y.Z. analyzed data. L.Z., S.S., A.J.A., and Y.Z. wrote the manuscript.

## Competing interests

The authors declare no competing interests.
