## [Peer Review file · Nature Communications]

A Versatile Platform for Chemical Engineering of Exosomes Empowered by ADP-Ribosyl Cyclases

Corresponding Author: Professor Yong Zhang

Version 0:

Reviewer comments:

Reviewer #1

(Remarks to the Author)

This study developed a CD38 enzyme-based chemically engineered platform. Leveraging the synergistic interplay between CD38's catalytic activity and the covalent inhibitor 2'-Cl-araNAD⁺, this platform achieves site-specific functionalization of exosome surfaces. The platform's versatility was systematically validated through the successful loading of diverse functional moieties, including: a fluorescent label (Alexa Fluor™ 488), a targeting ligand (Folic Acid, FA), a chemotherapeutic agent (Monomethyl auristatin F, MMAF), and a bone-targeting agent (Bisphosphonate, BP). The bioactivity and specificity of the functionalized exosomes were confirmed in both in vitro and in vivo models. Overall, this work represents a significant advancement in the development and innovation of exosome-based drug delivery platforms. However, prior to acceptance of this manuscript, the authors must address the following questions:

1. In Fig 2B, the authors detected the expression of fusion proteins constructed by Western Blot. The molecular weight should be labeled in the blot. Meanwhile, why there are two HA bands in CD38-PDGFR Exos? Why the CD9-CD38 showed the same size as CD9 as detected by CD9 antibody? The authors should provide more evidence to show the successful construction of these proteins as this is the foundation of this manuscript, such as sequence data as supplemental. Besides, the authors should as detect exosomal CD38 expression before and after transfection.
2. The level of exosomal CD38 should be quantified since this may result in the different catalytical rate as measured in Fig 2C.
3. Currently, only FA, MMAF, and BP are validated. It is recommended to supplement conjugation experiments with other types of molecules (such as antibodies and siRNA) to strengthen the universality of the platform.
4. The quantitative relationship between the expression level of CD38 enzyme on the surface of exosomes and functionalization efficiency needs to be further clarified, for example, by verifying the claim of "an average of 1,000 CD38/exosomes" through mass spectrometry or single-molecule counting. While the enzymatic activity of the CD9-CD38 fusion protein is significantly higher than that of CD38-PDGFR (Fig. 2C), the structure–function relationship has not been elucidated, particularly the impact of topological differences in the transmembrane domain on the catalytic center of CD38.
5. One of the advantages of this system different from other chemical modification methods is its homogenous expression between exosomes. The authors should verify this at the single exosome level.
6. In Fig 2D, why the native and CD9-CD38 Exos have different size distribution? Native Exos showed multiple peaks with several peaks greater than 200. In contrast, no large peaks were observed. Instead, only peaks less than 100 nm were detected.
7. Why there are also green fluorescence in other groups in Fig 2G?
8. In Fig 3, all the "concentration" was wrongly spelled as "concerntration".
9. In Page 13, line 216, the authors claimed Fig 4B was obtained under 4°C for 2h. However, in the corresponding figure legend part, the condition was 37°C for 1h. This inconsistency should be carefully checked throughout the manuscript.

10. HA was used for both hemagglutinin and hydroxyapatite.
11. In Fig 5H, the in vivo biodistribution should be performed in live whole animal at various timepoints. Meanwhile, the tumor cells should be labeled with Luciferase to show the location of the tumor to verify whether there is any colocalization with exosome administrated.
12. Instead of using a xenograft mouse model, all the in vivo experiments must be verified in an orthotopic mouse model with complete immune system. Meanwhile, why breast cancer cells, not bone cells, were used for bone study?
13. Why different animal numbers were used in Fig 7B.
14. In the in vivo evaluation, the authors should also perform more studies to verify the function of exosomes such as Goldner trichrome staining of femur to observe the mineralized bone, TRAP staining to assess the numbers of osteoclasts.
15. References should be cited after each specific statement, instead of citing multiple references together after a paragraph. For example, lines 42, 45, 47, 56, and 63.
16. The small molecule payloads mentioned in lines 64 to 70 of the manuscript, including Alexa 488 fluorophore, folic acid (FA) ligand, monomethyl auristatin F (MMAF) tubulin inhibitor, and bisphosphonate (BP) bone-targeting agent, are not displayed in the figure. It is unreasonable to cite Figure 1 here.
17. The purpose and significance of the fusion protein design described in lines 83–94 of the manuscript are not clearly stated. For example, the function and significance of the G4S linker are not mentioned. Revision is recommended. In addition, a schematic diagram of the fusion protein structure design would help to better illustrate the concept.
18. In the immunoblot analysis, why is there such a significant difference in the levels of CD9 and CD81 among the different groups of exosomes? Please provide an explanation.
19. Figure 5a suggests that the engineered exosomes designed in the manuscript are capable of targeting bones, tumors, and T cells simultaneously. However, the efficiency and proportion of exosome targeting to each of these cell types should be characterized and validated. Additionally, please clarify whether the exosomes can truly target multiple cell types simultaneously, or if they predominantly target only one specific cell type under the experimental conditions.
20. Due to the complex design and variety of modifications applied to the exosomes in the manuscript, please clearly define and standardize the nomenclature for each type of exosome based on their composition. The current naming system lacks clarity and may lead to confusion.
21. The usage of "Exo" and "exosome" should be standardized throughout the manuscript; for example, on line 311, where "Exo" is used without clear definition or consistency. Please ensure consistent nomenclature for clarity.
22. In the efficacy experiment, no control group with free MMAF or non-targeted MMAF-ARC Exos was included (Fig. 7), making it difficult to fully attribute the observed therapeutic effects to the advantage of targeted delivery. Please address this concern with appropriate experimental controls.
23. The synthesis of 2'-Cl-araNAD⁺-MMAF refers to previous work (Ref. 46); however, the key chemical steps involving the conjugation of MMAF to the inhibitor should be described in detail in the Supplementary Materials, including reaction yield and purity validation.

Reviewer #2

(Remarks to the Author)

This manuscript represents a significant advancement in the field of exosome-based therapeutics, showcasing a versatile platform for site-specific chemical engineering of exosomes through the enzymatic activity of surface-displayed CD38 and its covalent inhibitor, 2'-Cl-araNAD⁺. The ARC Exos system is effectively demonstrated to enable precise functionalization with small molecules, such as fluorescent dyes, receptor ligands, cytotoxic agents, and bone-targeting molecules, addressing challenges of heterogeneity and instability in traditional exosome engineering methods. The preclinical performance of BP- α CD3- α EGFR-ARC Exos, particularly in suppressing tumor progression and metastasis while boosting T-cell immunity, is compelling and highlights the platform's translational potential.

However, there are several points that require attention to further enhance the quality and clarity of the manuscript:

Title

1. The title of the paper is too general and does not fully reflect the innovative aspects of the proposed platform. A more specific title would improve clarity and impact.

Introduction

2. The authors frequently cite multiple references, often exceeding 10, within a single paragraph or for a single sentence without clearly explaining the relevance or justification for each citation. Clarifying the significance and relevance of references would not only highlight the authors' research logic but also enrich the content of the introduction section.

Results

3. The comparison between MMAF-ARC-Exo and MMAF-FA-ARC-Exo in Figure 3G is hard to interpret due to the lack of quantification of MMAF conjugation. The authors could assess the level of MMAF on exosomes using an anti-MMAF antibody. Additionally, MMAF-ARC-Exo appears to have a relatively high EC50, raising concerns about its potency. Figure 3D indicates that FA-ARC-exosomes largely remain on the cell surface, suggesting inefficient endocytosis. Quantifying uptake efficiency would provide valuable insights into the delivery mechanism and highlight potential limitations of the platform for endocytosis-dependent applications.
4. The stability of ARC Exos in serum should be investigated. The possibility that conjugated small molecules may be released by serum enzymes, such as through the pyrophosphate bond, particularly in the case of MMAF, should be assessed to estimate potential non-specific toxicities.
5. While CD38 enzymatic activity on exosomes has been confirmed (Figure 2C), the residual activity after reaction with 2'-Cl-araNAD⁺ and its conjugates was not assessed. Given that CD38 activity is known to contribute to immunosuppression, it is important to evaluate whether residual activity could influence T-cell function, particularly in the context of immunotherapy.
6. For the in vivo model, the BP molecule may nonspecifically target bone, potentially leading to side effects such as recruitment of T cells to normal bone tissue and disruption of normal bone function. The authors could address this toxicity concern by analyzing the structure of the right cancer-free hindlimb (Figure 7).
7. Molecular weight markers should be included in all Western blot figures to ensure clarity and reproducibility.
8. The contrast in several confocal microscopy images, such as those in Figs. 2G, 3D, and 7K, could be improved by reducing DAPI (blue) signal intensity and/or enhancing PKH67 (green) signals to improve visualization.

Discussion

9. CD38, as a multifunctional protein, functions not only as an enzyme but also as a receptor, ligand, and more. Therefore, the expression of CD38 on exosomes may impact other cell types that express its interacting proteins, such as CD31. Potential off-target effects should be addressed in the discussion.
10. The authors use glycosylation mutants of CD38 on exosomes, which may lead to potential immunogenicity issues. It would be helpful to compare the performance of wild-type CD38 in this context or discuss the potential impact of glycosylation loss on immune compatibility.
11. A brief discussion comparing the advantages and limitations of exosome-based delivery versus multispecific antibodies would be valuable for readers to assess the relative merits and future applications of the proposed approach. Additionally, the authors should discuss whether the properties of exosomes as a vehicle (e.g., different cellular origins) may influence the results.

Methods

12. Certain methodological details lack sufficient clarity, such as the preparation method for bone sections and the specific vein used for intravenous PBMC injections. Including these details would enhance reproducibility.
 13. For the in vitro experiments, the authors provide data for $n \geq 2$. However, $n \geq 3$ is generally required to ensure statistical robustness. The authors should ensure that all in vitro experiments meet this standard.
- Confusion in naming and abbreviation
14. There is confusion in the naming of the exosome constructs, such as "CD9-CD38" and "ARC," which appear to be used interchangeably within the same figure. Furthermore, constructs like "CD9-CD38- α CD3- α EGFR" are ambiguous in distinguishing which components are fused and which are co-expressed. The authors should adopt a consistent and unambiguous naming scheme, potentially utilizing colons or slashes (e.g., CD9-CD38/ α CD3- α EGFR) to clearly differentiate between fused and co-expressed components, ensuring clarity for readers.
 15. Another abbreviation that may cause confusion is "HA," which is used to refer to both hemagglutinin and hydroxyapatite. It is recommended to rename hydroxyapatite as "HAp" to avoid ambiguity.

Reviewer #3

(Remarks to the Author)

Reviewer #4

(Remarks to the Author)

The manuscript by Zhang et al includes some very detailed work describing novel methods for labelling exosomes. These methods will be of interest to the exosome community, and the manuscript is well written. The data are very clean and demonstrate the utility of this approach. While the in vivo results look very promising in terms of therapeutic efficacy, it is important to recognize that the use of NSG mice avoids problematic immune responses to the modified exosomes that would likely prohibit repeat injection and limit their clinical utility. That being said, the data are impressive and demonstrate the ability to conjugate multiple moieties to exosomes.

Version 1:

Reviewer comments:

Reviewer #1

(Remarks to the Author)

The authors have responded comprehensively and thoughtfully to the initial review comments. They have conducted extensive additional experiments and data analyses, including quantification of CD38 expression levels, bone staining assays, more in-depth functional validation both in vitro and in vivo, and improved image presentation. These supplementary data significantly enhance the scientific rigor of the manuscript and the credibility of its conclusions. The described ARC Exo platform technology for site-specific functionalization of exosomes is highly innovative and holds considerable application potential.

However, several aspects still require further clarification or exhibit shortcomings:

1. While the demonstration with small-molecule payloads is substantial, the claims of a "Versatile Platform" and a "general approach for chemical reprogramming exosomes" in the title and abstract appear somewhat overstated in the absence of supporting data for the conjugation of larger biomolecules such as proteins or nucleic acids. It is recommended to either temper such absolute statements in the discussion or provide preliminary proof-of-concept data (e.g., successful conjugation of a model protein or short peptide) to more robustly support the claim of "versatility".
2. The authors quantified CD38 expression by three methods and attributed the higher catalytic activity of CD9-CD38 exosomes to their higher CD38 expression levels rather than to potential effects of transmembrane topology differences on the catalytic center. While this explanation is reasonable, it remains somewhat superficial. The original question regarding the "impact of transmembrane domain topological differences on the CD38 catalytic center" touches upon a deeper mechanistic aspect. The authors could enhance the discussion by elaborating on why the CD9 fusion leads to substantially higher expression levels compared to the PDGFR-TMD fusion. Is it related to CD9's nature as an abundant exosomal tetraspanin, potentially facilitating more efficient recruitment of the fusion protein into the exosome biogenesis pathway? Incorporating discussion or relevant citations on this point would increase the manuscript's depth.
3. The authors provided a reasonable explanation for using the NSG mouse model engrafted with human PBMCs (targeting human CD3 and EGFR) and acknowledged the need for future evaluation in immunocompetent models. This represents an inherent limitation of the study. Although mentioned in the discussion, the implications of this model choice could be more thoroughly discussed, particularly concerning the assessment of repeat dosing feasibility and host immune responses. The immunogenicity risk associated with the modified exosomes (especially using deglycosylated CD38 mutants and synthetic chemical linkers) cannot be assessed in the NSG model. It is advisable to explicitly state in the discussion that the current excellent in vivo efficacy was achieved in a model circumventing allogeneic immune rejection, which presents a significant challenge for future clinical translation. If feasible, adding evaluation data in an immunocompetent mouse model would more comprehensively demonstrate the robustness of the design.
4. Regarding the previously mentioned size distribution differences in Fig. 2D, the authors' explanation, while reasonable, lacks statistical rigor and validation. It is recommended to perform replicate experiments to ensure the reproducibility and stability of the exosome preparations.
5. The response concerning the green fluorescence in other groups in Fig. 2G (attributed to background) seems somewhat insufficiently explored. The fluorescence intensity in the green channel of the confocal images appears potentially over-saturated. It is recommended to adjust the channel acquisition parameters and repeat the experiment to verify the accuracy of the observations.
6. In the newly added flow cytometry results (Fig. S10d), the positive percentages for BT-20 cells and T cells exceed 100%, which is inconsistent with experimental description. If PKH67-labeled BP- α CD3- α EGFR-ARC Exos were incubated with a mixture of BT-20 cells and T cells followed by flow cytometry, the percentages for BT-20 and T cells should be $\leq 100\%$. Furthermore, this experiment does not adequately validate the efficiency and proportion of exosomes targeting each cell type.
7. The authors demonstrated the bone-targeting capability of BP-ARC Exos through bone section binding assays and in vivo biodistribution. However, the possibility of non-specific binding of BP to other mineralized tissues (e.g., teeth) was not addressed. Additionally, the distribution ratio of exosomes between bone tumor tissue and normal bone tissue was not quantified.

In summary, the revisions have significantly improved the scientific quality and completeness of the manuscript.

Nevertheless, shortcomings remain regarding the platform's generality, the discussion of transmembrane domain effects, and the assessment of in vivo immunogenicity. Addressing the points above, particularly through additional experimental evidence where feasible, would further strengthen the study. If these concerns can be adequately addressed, the work would better meet the journal's publication standards.

Reviewer #2

(Remarks to the Author)

All of our concerns have been well addressed by the authors.

Reviewer #3

(Remarks to the Author)

Reviewer #4

(Remarks to the Author)

This is a very nice contribution.

Version 2:

Reviewer comments:

Reviewer #1

(Remarks to the Author)

I have no further comments

Dear reviewers,

Thank you for your positive responses to this work and for your critical and helpful comments. We have performed additional experiments and analysis and revised the manuscript and supporting information accordingly to address all issues you raised. A copy of the manuscript and supporting information with highlighted changes in red is enclosed for your review. Below please find the summarized point-by-point response.

Reviewer #1:

1. In Fig 2B, the authors detected the expression of fusion proteins constructed by Western Blot. The molecular weight should be labeled in the blot. Meanwhile, why there are two HA bands in CD38-PDGFR Exos? Why the CD9-CD38 showed the same size as CD9 as detected by CD9 antibody? The authors should provide more evidence to show the successful construction of these proteins as this is the foundation of this manuscript, such as sequence data as supplemental. Besides, the authors should as detect exosomal CD38 expression before and after transfection.

The authors appreciate this comment. Molecular weights for all immunoblot images in the revised manuscript have been included. For CD38-PDGFR Exos, the lower band matches the calculated weight (~ 38.6 kDa). The upper band seems to be the expressed fusion with an uncleaved N-terminal signal peptide or an unexpected posttranslational modification. For CD9-CD38 Exos, two bands were detected by the anti-CD9 antibody (see uncropped immunoblot below and in the Source Data file), which are CD9-CD38 fusion for the upper band and native CD9 for the lower band. They are presented separately in Fig. 2B. Amino acid sequences of the two fusion designs are now included in Table S1 in Supporting Information. Moreover, Immunoblots using an anti-human CD38 antibody were performed for the purified exosome samples and are now included in revised Fig. 2B. In contrast to CD38-PDGFR Exos and CD9-CD38 Exos (after transfection), native exosomes (before transfection) show little or no CD38 expression based on the anti-CD38 immunoblot.

2. *The level of exosomal CD38 should be quantified since this may result in the different catalytical rate as measured in Fig 2C.*

The authors appreciate this comment. The levels of exosomal CD38 for each fusion construct were quantified by three different methods, including NGD⁺-based enzymatic activity assays (1370 CD38 per CD9-CD38 Exo; 167 CD38 per CD38-PDGFR Exo), immunoblot (1215 CD38 per CD9-CD38 Exo; 195 CD38 per CD38-PDGFR Exo), and ELISA (1165 CD38 per CD9-CD38 Exo; 187 CD38 per CD38-PDGFR Exo). Purified recombinant CD38 extracellular domain was used as standards. These results shown in Fig. S1 in Supporting Information indicated increased levels of CD38 expression on CD9-CD38 Exos over those of CD38-PDGFR Exos, and a positive correlation between CD38 expression and its catalytic activity on exosomes.

3. *Currently, only FA, MMAF, and BP are validated. It is recommended to supplement conjugation experiments with other types of molecules (such as antibodies and siRNA) to strengthen the universality of the platform.*

The authors appreciate this comment. Several small-molecule model payloads were demonstrated for exosome conjugation by the reported ARC Exo technology. Other biomolecules such as antibodies and siRNAs can also be conjugated with exosomes through this technology. However, the relatively increased sizes for antibodies and siRNAs would require identification of optimal chemical modification sites and linker structures to ensure homogeneous and efficient conjugation without loss of biological functions, which will be explored in future studies. We have now discussed these candidates for conjugation in the first paragraph on page 30.

4. *The quantitative relationship between the expression level of CD38 enzyme on the surface of exosomes and functionalization efficiency needs to be further clarified, for example, by verifying the claim of “an average of 1,000 CD38/exosomes” through mass spectrometry or single-molecule counting. While the enzymatic activity of the CD9-CD38 fusion protein is significantly higher than that of CD38-PDGFR (Fig. 2C), the structure–function relationship has not been elucidated, particularly the impact of topological differences in the transmembrane domain on the catalytic center of CD38.*

The authors appreciate this comment. The expression levels of CD38 on exosomes were quantified by NGD⁺-based enzymatic activity assays, immunoblots, and ELISA assays. As shown in Fig. S1 in Supporting Information, it indicated significantly higher levels of CD38 expression on CD9-CD38 Exos than those of CD38-PDGFR Exos and a positive correlation between CD38 expression and its catalytic activity on exosomes. Based on these results, the catalytic rate difference between CD9-CD38 exosomes and CD38-PDGFR exosomes likely results from differential expression levels of CD38 fusion proteins on exosomes rather than activity variations for two fusion designs. These results are now described on page 7 and discussed in the second paragraph on page 30.

5. One of the advantages of this system different from other chemical modification methods is its homogenous expression between exosomes. The authors should verify this at the single exosome level.

The authors appreciate this comment. The homogenous expression was examined by nanoflow cytometric analysis of Alexa 488-ARC Exos at the single exosome level. As shown in Fig. S2f and S2g, fluorescence intensity distribution of Alexa 488-ARC Exos indicated homogeneous expression of the CD38 fusion and conjugation. Furthermore, compared with lipophilic PKH67 dye that randomly stains exosome membrane, nanoflow cytometry showed that the ARC Exo method can result in more efficient labeling of exosomes and little effects on exosome size distribution. These results are provided in Fig. S2f and S2g in Supporting Information and described on page 10.

6. In Fig 2D, why the native and CD9-CD38 Exos have different size distribution? Native Exos showed multiple peaks with several peaks greater than 200. In contrast, no large peaks were observed. Instead, only peaks less than 100 nm were detected.

The authors appreciate this comment. There are some variations in size distribution among different exosome constructs, which is commonly observed in our and other labs' studies. This is likely due to transfection that might slightly affect the generation of exosomes. Importantly, the primary peaks of these exosome constructs are all around 80-150 nm.

7. Why there are also green fluorescence in other groups in Fig 2G?

The authors appreciate this comment. Weak green fluorescence from other groups was background of confocal images, which might result from non-specific reactions of DBCO-Alexa 488 with exosomes and subsequent cellular uptake of these exosomes.

8. In Fig 3, all the "concentration" was wrongly spelled as "concerntration".

The authors appreciate this comment and apologize for this typo. The labels in Fig. 3 have been corrected.

9. In Page 13, line 216, the authors claimed Fig 4B was obtained under 4°C for 2h. However, in the corresponding figure legend part, the condition was 37°C for 1h. This inconsistency should be carefully checked throughout the manuscript.

The authors appreciate this comment and apologize for the inconsistent information. It should be 37°C for 1 h. This is now corrected on page 13.

10. HA was used for both hemagglutinin and hydroxyapatite.

The authors appreciate this comment. To distinguish them, hemagglutinin is still abbreviated as HA, but hydroxyapatite is now abbreviated as HAp. All related figures and texts have been revised.

11. In Fig 5H, the in vivo biodistribution should be performed in live whole animal at various timepoints. Meanwhile, the tumor cells should be labeled with Luciferase to show the location of the tumor to verify whether there is any colocalization with exosome administrated.

The authors appreciate this comment. Luminescence signals from luciferase-labeled tumors and fluorescence signals from DiR-labeled exosomes were collected at varied time points before euthanizing mice at the end of the biodistribution study. Representative images of whole animals were added in Fig. S7b. Luciferase signal-positive areas were circled with red dotted lines. Significant co-localization was observed for tumor-derived luminescence and fluorescent signals of exosomes. This is also described on page 16.

12. Instead of using a xenograft mouse model, all the in vivo experiments must be verified in an orthotopic mouse model with complete immune system. Meanwhile, why breast cancer cells, not bone cells, were used for bone study?

The authors appreciate this comment. NSG mouse xenograft models with grafted human PBMCs were used for *in vivo* assessments of anti-tumor immunity of the developed BP- α CD3- α EGFR-ARC Exos, because BP- α CD3- α EGFR-ARC Exos target human T cells and EGFR expressed on human cancer cells and do not cross-react with mouse CD3 or EGFR antigens. The presented two NSG mice-based studies demonstrated that conjugated BP molecules on exosomes via the ARC Exo approach can promote immunity against tumors formed on bones in this proof-of-concept study. To comprehensively characterize immune responses induced by the reported form of ARC Exos, syngeneic mouse models along with ARC Exos specific for mouse antigens will be developed for future studies. These are now discussed in the last paragraph of the Discussion section on page 31.

As bone is a common site for breast cancer metastasis and triple negative breast cancer cell lines BT20 and MDA-MB-231 express considerable levels of EGFR, these two mouse xenograft models were selected for evaluating efficacy and specificity of the bone-targeted BP- α CD3- α EGFR-ARC Exos. Importantly, bone cells could be selected for future studies with BP-ARC Exos or other bone-specific ARC Exos recognizing bone cell-associated antigens.

13. Why different animal numbers were used in Fig 7B.

The authors appreciate this comment. Based on our pilot animal studies, tumors from the BP- α CD3- α EGFR-ARC Exos-treated group tend to be tiny or difficult for detection. To ensure tumor tissues could be collected from at least five mice for T-cell infiltration analysis at the end of the study, additional mice were included for the BP- α CD3- α EGFR-ARC Exos group.

14. In the in vivo evaluation, the authors should also perform more studies to verify the function of exosomes such as Goldner trichrome staining of femur to observe the mineralized bone, TRAP staining to assess the numbers of osteoclasts.

The authors appreciate this comment. Masson-Goldner trichrome and TRAP staining were performed for both animal studies. Results are now included in Fig. S16, S21b, and S22a in Supporting Information and described on pages 22 and 26. Compared with BP- α CD3- α EGFR-ARC Exos-treated mice, increased losses of muscle and bone tissues and numbers of osteoclasts were observed for mice from other groups.

15. References should be cited after each specific statement, instead of citing multiple references together after a paragraph. For example, lines 42, 45, 47, 56, and 63.

The authors appreciate this comment. References are now cited after each specific statement.

16. The small molecule payloads mentioned in lines 64 to 70 of the manuscript, including Alexa 488 fluorophore, folic acid (FA) ligand, monomethyl auristatin F (MMAF) tubulin inhibitor, and bisphosphonate (BP) bone-targeting agent, are not displayed in the figure. It is unreasonable to cite Figure 1 here.

The authors appreciate this comment. Citation for Fig. 1 after this sentence is now removed.

17. The purpose and significance of the fusion protein design described in lines 83–94 of the manuscript are not clearly stated. For example, the function and significance of the G4S linker are not mentioned. Revision is recommended. In addition, a schematic diagram of the fusion protein structure design would help to better illustrate the concept.

The authors appreciate this comment. To utilize the type I PDGFR TMD for exosomal display, CD38 extracellular domain was placed at its N-terminus (Fig. 2A). As CD9 is abundant in exosomes and has no extracellular termini, full-length CD38 containing the cytoplasmic domain and TMD was fused to the C-terminus of CD9. To improve folding of the designed fusion proteins as well as separate the CD38 domain from the fusion partner, a flexible G₄S or (G₄S)₂ linker was inserted between them. The design and rationale of these fusion proteins are now further described on page 6. In addition, the domain map for each fusion design is included in Table S1 in Supporting Information.

18. In the immunoblot analysis, why is there such a significant difference in the levels of CD9 and CD81 among the different groups of exosomes? Please provide an explanation.

The authors appreciate this comment. The variation in expression levels of CD9 and CD81 among different groups are likely due to transfection of expression vectors and/or exosomal surface display of designed CD38 fusion proteins. The transfection process could alter the cell physiological state and disturb expression levels and subcellular locations of CD9 and CD81. Also, expression of the CD38-PDGFR TMD and CD9-CD38 fusion proteins on membranes could affect expression and distribution of native CD9 and CD81 proteins.

19. Figure 5a suggests that the engineered exosomes designed in the manuscript are capable of targeting bones, tumors, and T cells simultaneously. However, the efficiency and proportion of exosome targeting to each of these cell types should be characterized and validated. Additionally, please clarify whether the exosomes can truly target multiple cell types simultaneously, or if they predominantly target only one specific cell type under the experimental conditions.

The authors appreciate this comment. To demonstrate BP- α CD3- α EGFR-ARC Exos can simultaneously bind to T cells, EGFR-positive cancer cells, and bone tissues, native exosomes, CD9-CD38- α CD3- α EGFR Exos, and BP- α CD3- α EGFR-ARC Exos were labeled with PKH67 fluorescent dyes for confocal microscopic studies with mixtures of T cells, BT-20 cells, bone sections. As shown in Fig. S10a in Supporting Information, unlike native exosomes and CD9-CD38- α CD3- α EGFR Exos, BP- α CD3- α EGFR-ARC Exos promote the clustering of BT-20 cells with T cells on bone tissues.

Moreover, to quantify binding efficiency and proportion of BP- α CD3- α EGFR-ARC Exos, mixtures of BT-20 and T cells were incubated with PKH67 labeled BP- α CD3- α EGFR-ARC Exos for 0-60 min, followed by flow cytometry analysis. As shown in Fig. S10b-S10e in Supporting Information, BP- α CD3- α EGFR-ARC Exos could bind to most BT-20 cells and T cells within 15-30 min. The amounts of BP- α CD3- α EGFR-ARC Exos on/in cells are time-dependent. The binding proportions of BP- α CD3- α EGFR-ARC Exos to each cell type seems to be dependent on expression levels of CD3 on T cells and EGFR on BT-20 cells. In addition to the added Fig. S10 in Supporting Information, these results are described on page 19.

20. Due to the complex design and variety of modifications applied to the exosomes in the manuscript, please clearly define and standardize the nomenclature for each type of exosome based on their composition. The current naming system lacks clarity and may lead to confusion.

The authors appreciate this comment. The ARC Exo is defined as exosome expressing the CD9-CD38 fusion with conjugated functional groups (e.g. Alexa 488-ARC Exo has Alexa 488 conjugated to the expressed CD9-CD38 fusion; FA-ARC Exo carries FA conjugated to the CD9-CD38 fusion; BP-ARC Exo includes BP conjugated to the CD9-CD38 fusion; and BP- α CD3- α EGFR-ARC Exo contains BP conjugated to the CD9-CD38 fusion and the expressed α CD3- α EGFR-PDGFR TMD fusion).

For non-conjugated exosomes, they are named based on expressed fusion proteins (e.g. CD9-CD38 exosome expressing the CD9-CD38 fusion; α CD3- α EGFR Exo expressing α CD3- α EGFR-PDGFR TMD fusion). To clearly indicate the composition, "CD9-CD38- α CD3- α EGFR Exos" is now changed to "CD9-CD38/ α CD3- α EGFR Exos" for reflection of two co-transfected plasmids encoding the CD9-CD38 fusion and α CD3- α EGFR-PDGFR TMD fusion.

21. The usage of "Exo" and "exosome" should be standardized throughout the manuscript; for example, on line 311, where "Exo" is used without clear definition or consistency. Please ensure consistent nomenclature for clarity.

The authors appreciate this comment. ARC Exos are defined on page 4. CD9-CD38/ α CD3- α EGFR Exos are now defined on page 15. α CD3- α EGFR Exos are now defined on page 16.

22. *In the efficacy experiment, no control group with free MMAF or non-targeted MMAF-ARC Exos was included (Fig. 7), making it difficult to fully attribute the observed therapeutic effects to the advantage of targeted delivery. Please address this concern with appropriate experimental controls.*

The authors appreciate this comment. To clarify, this *in vivo* study was to examine efficacy, specificity, and toxicity of BP- α CD3- α EGFR-ARC Exos which carried no MMAF payloads. BP- α CD3- α EGFR-ARC Exos were developed to redirect and activate cytotoxic T cells toward killing EGFR-positive tumor cells on bones, inducing cellular immunity against tumors on bones. No MMAF was loaded into BP- α CD3- α EGFR-ARC Exo for targeted delivery. All other relevant control groups were included for this study, which were PBS vehicle, native exosomes, BP-ARC Exos without the α CD3 and α EGFR scFv antibodies on surface, α CD3- α EGFR Exos without conjugated BP molecules, the combination of BP-ARC Exos and α CD3- α EGFR Exos, and CD9-CD38/ α CD3- α EGFR Exos expressing both fusion proteins but lacking conjugated BP molecules.

23. *The synthesis of 2'-Cl-araNAD⁺-MMAF refers to previous work (Ref. 46); however, the key chemical steps involving the conjugation of MMAF to the inhibitor should be described in detail in the Supplementary Materials, including reaction yield and purity validation.*

The authors appreciate this comment. Experimental details for synthesizing 2'-Cl-araNAD⁺-MMAF along with yields and purity are now included on page S46 in Supporting Information.

Reviewer #2:

1. *The title of the paper is too general and does not fully reflect the innovative aspects of the proposed platform. A more specific title would improve clarity and impact.*

The authors appreciate this comment. The title has been revised accordingly.

2. *The authors frequently cite multiple references, often exceeding 10, within a single paragraph or for a single sentence without clearly explaining the relevance or justification for each citation. Clarifying the significance and relevance of references would not only highlight the authors' research logic but also enrich the content of the introduction section.*

The authors appreciate this comment. References are now cited after each specific statement.

3. *The comparison between MMAF-ARC-Exo and MMAF-FA-ARC-Exo in Figure 3G is hard to interpret due to the lack of quantification of MMAF conjugation. The authors could assess the level of MMAF on exosomes using an anti-MMAF antibody. Additionally, MMAF-ARC-Exo appears to have a relatively high EC₅₀, raising concerns about its potency. Figure 3D indicates that FA-ARC-exosomes largely remain on the cell surface, suggesting inefficient endocytosis. Quantifying uptake efficiency would provide valuable insights into the delivery mechanism and highlight potential limitations of the platform for endocytosis-dependent applications.*

The authors appreciate this comment. MMAF levels on MMAF-ARC-Exos and FA-MMAF-ARC-Exos were quantified by ELISA assays using an anti-MMAF antibody and MMAF-conjugated

CD38 as standards. For MMAF-ARC Exos, there are about 950 MMAF molecules on each particle. For FA-MMAF-ARC Exos, around 490 MMAF molecules were conjugated on each vesicle. These results are included in Fig. S4a and S4b in Supporting Information and described on page 11.

The relatively high EC_{50} values for MMAF-ARC Exos likely result from inefficient cellular uptake and/or payload release in the absence of promotion by $FR\alpha$. To clarify, Fig. 3D shows the binding of FA-ARC Exos to $FR\alpha$ -positive cells at 4°C. Uptake efficiency of MMAF-ARC-Exos and FA-MMAF-ARC-Exos for these cell lines were assessed at 37°C. As shown in Fig. S3e-S3h in Supporting Information, FA-MMAF-ARC Exos exhibit significantly increased uptake efficiency for $FR\alpha$ -positive SKOV-3 and A2780 cells in comparison to MMAF-ARC Exos, but not for $FR\alpha$ -negative CAKI-1 and A549 cells. These results are also described on page 11.

4. The stability of ARC Exos in serum should be investigated. The possibility that conjugated small molecules may be released by serum enzymes, such as through the pyrophosphate bond, particularly in the case of MMAF, should be assessed to estimate potential non-specific toxicities.

The authors appreciate this comment. Stability of MMAF-ARC Exos in mouse plasma was examined at 37°C. The half-life of MMAF-ARC Exos is around 43.5 hour. These results are included in Fig. S4c and S4d in Supporting Information and described on pages 11 and 12.

5. While CD38 enzymatic activity on exosomes has been confirmed (Figure 2C), the residual activity after reaction with 2'-Cl-araNAD⁺ and its conjugates was not assessed. Given that CD38 activity is known to contribute to immunosuppression, it is important to evaluate whether residual activity could influence T-cell function, particularly in the context of immunotherapy.

The authors appreciate this comment. All exosome conjugation reactions with 2'-Cl-araNAD⁺ and its synthetic analogues were assessed via NGD⁺-based activity assays, which showed no detectable CD38 residual activities. These results are now included in Fig. S2e, S3a-S3c, S4a, S5, and S7a in Supporting Information.

To evaluate whether residual CD38 activity could affect T-cell function, PBMC viability and T-cell activation were examined in the presence of catalytically active CD9-CD38 exosomes at various concentrations. CD9-CD38 exosomes revealed little or weak effects on PBMC viability and T-cell activation. These results are included in Fig. S12 in Supporting Information and described on page 20.

6. For the in vivo model, the BP molecule may nonspecifically target bone, potentially leading to side effects such as recruitment of T cells to normal bone tissue and disruption of normal bone function. The authors could address this toxicity concern by analyzing the structure of the right cancer-free hindlimb (Figure 7).

The authors appreciate this comment. The right tumor-free hindlimbs from all groups in the BT-20 animal model study were scanned by micro-CT. Bone structures from micro-CT images indicated lack of noticeable toxicities. Those results are included in Fig. S17b in Supporting Information and described on page 22.

7. Molecular weight markers should be included in all Western blot figures to ensure clarity and reproducibility.

The authors appreciate this comment. Molecular weight markers are now included for all immunoblots in the revised manuscript.

8. The contrast in several confocal microscopy images, such as those in Figs. 2G, 3D, and 7K, could be improved by reducing DAPI (blue) signal intensity and/or enhancing PKH67 (green) signals to improve visualization.

The authors appreciate this comment. Confocal microscopy images with improved contrast are included in Fig. 2G, 3D, and 7K.

9. CD38, as a multifunctional protein, functions not only as an enzyme but also as a receptor, ligand, and more. Therefore, the expression of CD38 on exosomes may impact other cell types that express its interacting proteins, such as CD31. Potential off-target effects should be addressed in the discussion.

The authors appreciate this comment. In addition to functioning as an ectoenzyme, CD38 binds to several proteins. The potential specificity and toxicity issues for CD38-expressing exosomes are now discussed on page 31.

10. The authors use glycosylation mutants of CD38 on exosomes, which may lead to potential immunogenicity issues. It would be helpful to compare the performance of wild-type CD38 in this context or discuss the potential impact of glycosylation loss on immune compatibility.

The authors appreciate this comment. CD38 with four mutated asparagine residues for eliminating glycosylation was used for exosomal display. The potential immunogenicity issues for exosomes with mutated CD38 are now discussed on page 31.

11. A brief discussion comparing the advantages and limitations of exosome-based delivery versus multispecific antibodies would be valuable for readers to assess the relative merits and future applications of the proposed approach. Additionally, the authors should discuss whether the properties of exosomes as a vehicle (e.g., different cellular origins) may influence the results.

The authors appreciate this comment. The advantages and limitations of exosomes with displayed biomolecules compared with multi-specific antibodies as well as the effects of exosome composition on its physicochemical properties and biological activities are discussed on page 31.

12. Certain methodological details lack sufficient clarity, such as the preparation method for bone sections and the specific vein used for intravenous PBMC injections. Including these details would enhance reproducibility.

The authors appreciate this comment. Details for preparing bone sections are now included as a subsection of Preparation of non-decalcified bone sections in the Materials and Methods section on pages 39 and 40. Information about the vein used for PBMC injections was also added in the

Materials and Methods section on pages 46 and 47.

13. For the in vitro experiments, the authors provide data for $n \geq 2$. However, $n \geq 3$ is generally required to ensure statistical robustness. The authors should ensure that all in vitro experiments meet this standard.

The authors appreciate this comment. All *in vitro* experiments in the revised manuscript were performed with $n \geq 3$. This information in the subsection of Statistical analysis in the Materials and Methods section on page 49 has been updated accordingly.

14. There is confusion in the naming of the exosome constructs, such as “CD9-CD38” and “ARC,” which appear to be used interchangeably within the same figure. Furthermore, constructs like “CD9-CD38- α CD3- α EGFR” are ambiguous in distinguishing which components are fused, and which are co-expressed. The authors should adopt a consistent and unambiguous naming scheme, potentially utilizing colons or slashes (e.g., CD9-CD38/ α CD3- α EGFR) to clearly differentiate between fused and co-expressed components, ensuring clarity for readers.

The authors appreciate this comment. The ARC Exo is defined as exosome expressing the CD9-CD38 fusion with conjugated functional groups (e.g. Alexa 488-ARC Exo has Alexa 488 conjugated to the expressed CD9-CD38 fusion; FA-ARC Exo carries FA conjugated to the CD9-CD38 fusion; BP-ARC Exo includes BP conjugated to the CD9-CD38 fusion; and BP- α CD3- α EGFR-ARC Exo contains BP conjugated to the CD9-CD38 fusion and the expressed α CD3- α EGFR-PDGFR TMD fusion).

For non-conjugated exosomes, they are named based on expressed fusion proteins (e.g. CD9-CD38 exosome expressing the CD9-CD38 fusion; α CD3- α EGFR Exo expressing α CD3- α EGFR-PDGFR TMD fusion). To clearly indicate the composition (i.e. fusion or co-expression), “CD9-CD38- α CD3- α EGFR Exos” is now changed to “CD9-CD38/ α CD3- α EGFR Exos” for reflection of two co-transfected plasmids encoding the CD9-CD38 fusion and α CD3- α EGFR-PDGFR TMD fusion.

15. Another abbreviation that may cause confusion is “HA,” which is used to refer to both hemagglutinin and hydroxyapatite. It is recommended to rename hydroxyapatite as “HAp” to avoid ambiguity.

The authors appreciate this comment. Hydroxyapatite is now abbreviated as HAp. All related figures and texts have been revised accordingly.

Reviewer #3:

The authors appreciate this reviewer’s valuable comments.

Reviewer #4:

The manuscript by Zhang et al includes some very detailed work describing novel methods for labelling exosomes. These methods will be of interest to the exosome community, and the manuscript is well written. The data are very clean and demonstrate the utility of this approach. While the in vivo results look very promising in terms of therapeutic efficacy, it is important to recognize that the use of NSG mice avoids problematic immune responses to the modified exosomes that would likely prohibit repeat injection and limit their clinical utility. That being said, the data are impressive and demonstrate the ability to conjugate multiple moieties to exosomes.

The authors appreciate this reviewer's insightful comments. NSG mouse xenograft models with grafted human PBMCs were used for *in vivo* assessments of anti-tumor immunity of the developed BP- α CD3- α EGFR-ARC Exos, because BP- α CD3- α EGFR-ARC Exos target human T cells and EGFR expressed on human cancer cells and do not cross-react with mouse CD3 or EGFR antigens. The conjugated functional groups on ARC Exos may trigger host immune responses. To evaluate potential immunogenicity of ARC Exos, immunocompetent mice will be used for future studies. These are now discussed on page 31.

Thank you for your time and efforts.

Dear reviewer,

Thank you for your positive responses to this work and for your critical and helpful comments. We have performed additional experiments and analysis and revised the manuscript and supporting information accordingly to address all issues you raised. A copy of the manuscript and supporting information with highlighted changes in red is enclosed for your review. Below please find the summarized point-by-point response.

Reviewer #1:

1. While the demonstration with small-molecule payloads is substantial, the claims of a "Versatile Platform" and a "general approach for chemical reprogramming exosomes" in the title and abstract appear somewhat overstated in the absence of supporting data for the conjugation of larger biomolecules such as proteins or nucleic acids. It is recommended to either temper such absolute statements in the discussion or provide preliminary proof-of-concept data (e.g., successful conjugation of a model protein or short peptide) to more robustly support the claim of "versatility".

The authors appreciate this comment. Among several payloads that were shown to be successfully conjugated with exosomes in this study is MMAF, a peptide analogue with five amino acids. For other proteins and nucleic acids with increased sizes, chemical modification sites and linker structures would need to be explored and optimized in future studies to ensure homogeneous and efficient conjugation without loss of activities. Given the lack of supporting data for conjugating larger biomolecules, we have accordingly revised the first, second, and fourth paragraph in the Discussion section on pages 29-31.

2. The authors quantified CD38 expression by three methods and attributed the higher catalytic activity of CD9-CD38 exosomes to their higher CD38 expression levels rather than to potential effects of transmembrane topology differences on the catalytic center. While this explanation is reasonable, it remains somewhat superficial. The original question regarding the "impact of transmembrane domain topological differences on the CD38 catalytic center" touches upon a deeper mechanistic aspect. The authors could enhance the discussion by elaborating on why the CD9 fusion leads to substantially higher expression levels compared to the PDGFR-TMD fusion. Is it related to CD9's nature as an abundant exosomal tetraspanin, potentially facilitating more efficient recruitment of the fusion protein into the exosome biogenesis pathway? Incorporating discussion or relevant citations on this point would increase the manuscript's depth.

The authors appreciate this comment. Our quantitative analysis results suggest that the catalytic rate difference between CD9-CD38 exosomes and CD38-PDGFR exosomes are likely due to differential

expression levels of CD38 fusion proteins, because CD9 tetraspanin is highly abundant in exosomes in comparison with the single-pass PDGFR TMD. The enzymatic activities of CD38-PDGFR TMD and CD9-CD38 fusion proteins could also be different. The impact of transmembrane domain orientation on the fused CD38 activity can be assessed using appropriate model systems. These are now further discussed in the third paragraph of the Discussion section on page 30 and new references #63-65 regarding high abundance of CD9 on exosomes are included.

3. The authors provided a reasonable explanation for using the NSG mouse model engrafted with human PBMCs (targeting human CD3 and EGFR) and acknowledged the need for future evaluation in immunocompetent models. This represents an inherent limitation of the study. Although mentioned in the discussion, the implications of this model choice could be more thoroughly discussed, particularly concerning the assessment of repeat dosing feasibility and host immune responses. The immunogenicity risk associated with the modified exosomes (especially using deglycosylated CD38 mutants and synthetic chemical linkers) cannot be assessed in the NSG model. It is advisable to explicitly state in the discussion that the current excellent in vivo efficacy was achieved in a model circumventing allogeneic immune rejection, which presents a significant challenge for future clinical translation. If feasible, adding evaluation data in an immunocompetent mouse model would more comprehensively demonstrate the robustness of the design.

The authors appreciate this comment. In the fifth paragraph of the Discussion section on page 32, we further discussed that while the NSG model facilitates evaluation of *in vivo* efficacy of the reported ARC Exos, it prevents assessment of host immune responses and feasibility of repeated administration of these exosomes. Immunocompetent mice along with mouse cell-derived ARC Exos would need to be used for comprehensive characterization of their immunoreactivities in future studies.

4. Regarding the previously mentioned size distribution differences in Fig. 2D, the authors' explanation, while reasonable, lacks statistical rigor and validation. It is recommended to perform replicate experiments to ensure the reproducibility and stability of the exosome preparations.

The authors appreciate this comment. We repeated exosome preparations and NTA analysis. The new data are shown below with areas in red representing error bars. The size distributions of native exosomes and CD9-CD38 exosomes are comparable to those shown in Fig. 2D, peaking around 100 nm.

5. The response concerning the green fluorescence in other groups in Fig. 2G (attributed to background) seems somewhat insufficiently explored. The fluorescence intensity in the green channel of the confocal images appears potentially over-saturated. It is recommended to adjust the channel acquisition parameters and repeat the experiment to verify the accuracy of the observations.

The authors appreciate this comment. We have adjusted channel acquisition parameters and updated Fig. 2G in the revised manuscript. In addition, we repeated this experiment. The new data shown below are consistent with ones in Fig. 2G.

6. In the newly added flow cytometry results (Fig. S10d), the positive percentages for BT-20 cells and T cells exceed 100%, which is inconsistent with experimental description. If PKH67-labeled BP- α CD3- α EGFR-ARC Exos were incubated with a mixture of BT-20 cells and T cells followed by flow cytometry, the percentages for BT-20 and T cells should be $\leq 100\%$. Furthermore, this experiment does not adequately validate the efficiency and proportion of exosomes targeting each cell type.

The authors appreciate this comment. The BP- α CD3- α EGFR-ARC Exo-positive BT-20 cells and T cells reached approximately 98% and 90% at 60 min, respectively. To more adequately evaluate the efficiency and proportion of exosomes targeting each cell type, we analyzed cells via flow cytometry after incubation with BP- α CD3- α EGFR-ARC Exos for 15, 30, 60, 120, and 240 min, followed by quantitative analysis to calculate the binding efficiency and proportion of exosomes for each type of cells. The binding efficiency was determined as the percentage of cell with BP- α CD3- α EGFR-ARC Exos over time. The proportion of exosomes targeting each type of cells was quantified as the percentage of BP- α CD3- α EGFR-ARC Exos with each cell type at each time point. These new results are now included in Fig. S10b-S10e in Supporting Information.

7. The authors demonstrated the bone-targeting capability of BP-ARC Exos through bone section binding assays and in vivo biodistribution. However, the possibility of non-specific binding of BP to other mineralized tissues (e.g., teeth) was not addressed. Additionally, the distribution ratio of exosomes between bone tumor tissue and normal bone tissue was not quantified.

The authors appreciate this comment. We performed IVIS analysis for the teeth from mice in Fig. 5H. Only very low levels of BP- α CD3- α EGFR-ARC Exos signals were detected at 24 h, but not at 48, 96, and 168 h. The distribution ratio of BP- α CD3- α EGFR-ARC Exos for bone tumor tissue to normal bone tissue was also quantified. These new results are now included in Fig. S7b and S7d in Supporting Information and described in the Results section on page 16.

In summary, the revisions have significantly improved the scientific quality and completeness of the manuscript. Nevertheless, shortcomings remain regarding the platform's generality, the discussion of transmembrane domain effects, and the assessment of in vivo immunogenicity. Addressing the points above, particularly through additional experimental evidence where feasible, would further strengthen the study. If these concerns can be adequately addressed, the work would better meet the journal's publication standards.

The authors appreciate these comments. Above please find our point-by-point response.

Thank you for your time and efforts.